# Normalizing Flows are Capable Models for Continuous Control

**Raj Ghugare**     **Benjamin Eysenbach**
Department of Computer Science
Princeton University
rg9360@princeton.edu

## Abstract

Modern reinforcement learning (RL) algorithms have found success by using probabilistic models, such as transformers, energy-based models, and diffusion/flow-based models. To this end, researchers often choose to pay the price of accommodating these models into their algorithms – diffusion models are expressive, but are computationally intensive due to their reliance on solving differential equations, while autoregressive transformer models are scalable but typically require learning discrete representations. Normalizing flows (NFs), by contrast, seem to provide an appealing alternative, as they enable likelihoods and sampling without solving differential equations or autoregressive architectures. However, their potential in RL has received limited attention, partly due to the prevailing belief that normalizing flows lack sufficient expressivity. We show that this is not the case. Building on recent work in NFs, we propose a single NF architecture which integrates seamlessly into RL algorithms, serving as a policy, Q-function, and occupancy measure. Our approach leads to much simpler algorithms, and achieves higher performance in imitation learning, offline, goal conditioned RL and unsupervised RL.[1]

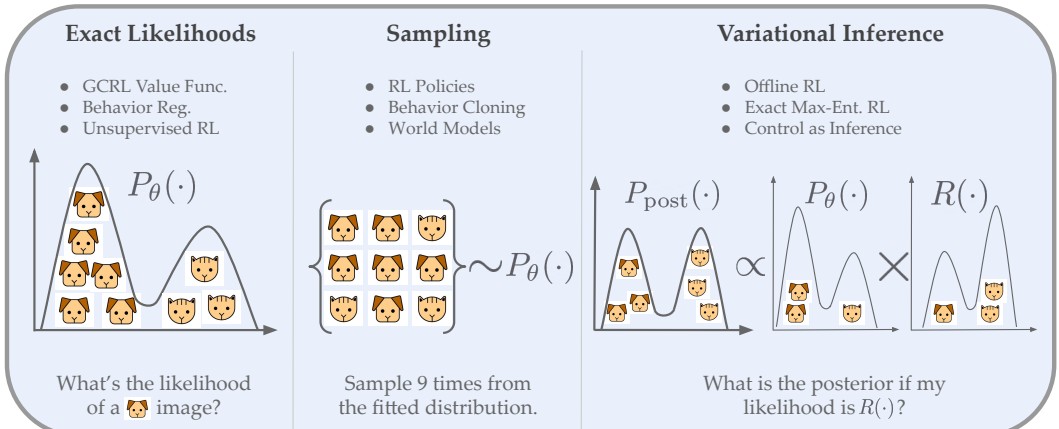

Figure 1: **Probabilistic models and RL.** Three capabilities of probabilistic models are important for RL: efficient likelihood computation, sampling, and compatibility with variational inference (VI). These capabilities are at the core of most RL algorithms (See Section 4 for details). For example, behavior cloning requires maximizing likelihoods and sampling [4], GCRL value functions require estimating future discounted state probabilities [23], and MaxEnt RL and RL finetuning requires variational inference [54]. However, models like diffusion and autoregressive transformers used in RL today support only a subset of these capabilities, making their application complex. In this paper, we show how normalizing flows, which posses these properties, can be a generally capable model family for RL.

---

[1]The code for all experiments can be found here : https://github.com/Princeton-RL/normalising-flows-4-reinforcement-learning.

39th Conference on Neural Information Processing Systems (NeurIPS 2025).

Table 1: **Capabilities of probabilistic models.** NFs have capabilities that make them applicable to a large number of RL problems (Section 4), including the ability to compute exact likelihoods, sample from complex distributions, and be trained via both maximum likelihood and variational inference (Section 3). Despite these capabilities, research on NFs in RL has been limited compared to other state of the art models. Is this because NFs lack expressivity? Our experiments (Section 6) provide compelling evidence that this is not the case.

| Model family | Exact Likelihoods | Sampling | Variational Inference[*] |
|---|---|---|---|
| Variational Autoencoders | Lower bound | Yes | No |
| Generative Adversarial networks | No | Yes | No |
| Energy Based models | Unnormalized | MCMC | No |
| Diffusion models | Yes, via ODEs | Unrolling SDEs | No |
| Normalizing Flows | Yes | Yes | Yes |

[*] Although directly performing VI (posterior sampling and estimation) is generally intractable for the first four model families we have still marked them with yellow because additional algorithms for VI do exist [8].

# 1 Introduction

Reinforcement learning (RL) algorithms often require probabilistic modeling of high dimensional and complex distributions. For example, offline RL or behavior cloning (BC) from large scale robotics datasets requires learning a multi-modal policy [4, 16, 7] and value function estimation in goal-conditioned RL (GCRL) requires estimating probabilities over future goals [23]. Different algorithms require different capabilities, including likelihood estimation, variational inference (VI) and efficient sampling (see Figure 1). A model trained via BC must maximize action likelihoods during training and support fast action sampling for real-time control. Most offline or finetuning RL algorithms require policy models to be able to backpropagate gradients of the value function through sampled actions. For MaxEnt RL with continuous actions, models should be optimizable using variational inference. However, obtaining these abilities in current models often comes at the cost of compute and complexity (Table 1). For example, diffusion models [78] are highly expressive yet computationally demanding: training, sampling, and evaluating log likelihoods all require solving a differential equation. Autoregressive transformer models [86] are scalable and expressive, but are typically limited to discrete settings; using them for RL in continuous spaces requires learning discrete representations of actions [76] or states [60]. Gaussian models offer computational efficiency but lack expressivity. *Can we design expressive probabilistic models with all the capabilities in Fig. 1?*

Normalizing Flows (NFs) are among the most flexible probabilistic models: they (1) support both likelihood maximization and variational inference training, and (2) enable efficient sampling and exact likelihood computation (see Table 1). Yet, NFs have received far less attention from the RL community, perhaps due to the (mis)conception that they have restricted architectures or training instabilities [65]. This paper demonstrates that neither of these notions hold NFs back in practice: NFs can match the expressivity of diffusion and autoregressive models while being significantly simpler. Moreover, unlike other models, NFs do not require additional machinery to be integrated with existing RL frameworks, enabling simpler algorithms that are competitive with if not better than their counterparts.

Our primary contribution is to shed light on the effectiveness and NFs in RL. Building on prior work [46, 20], we develop a simple NF architecture and evaluate it across 82 tasks spanning five different RL settings, including offline imitation learning, offline RL, goal-conditioned RL, and unsupervised RL:

- On offline imitation learning, NFs are competitive with baselines using state of the art generative models such as diffusion and transformers, while using significantly less compute ($6\times$ fewer parameters than diffusion policies).

- On offline conditional imitation learning, NFs outperform BC baselines using models like flow matching, and offline RL algorithms which learn a value function, across 45 tasks.

- On offline RL problems, NFs are competitive with strong baselines using diffusion / flow matching models on most tasks, outperforming all of them on 15/30 tasks.

- On unsupervised GCRL, NFs can rival contrastive learning based algorithms, despite making fewer assumptions.

## 2 Related Work

**Normalizing flows.** This work builds on the rich literature of normalizing flow architectures [19, 74, 46, 47, 40, 66, 91]. NICE [19] introduce the idea of stacking neural function blocks that perform shifting, making density estimation efficient. RealNVP [20] extends NICE to non volume preserving layers that also perform a scaling operation. GLOW [46] further propose applying an invertible generalized permutation and a special normalization layer (ActNorm) to improve expressivity.

**Probabilistic models in RL.** Strides in probabilistic modeling research consistently push the boundaries in deep RL. Variational autoencoders gave rise to a family of model-based [35, 36, 93] and representation learning [52, 90] algorithms. Contrastive learning has been used to learn value functions [59, 75, 25, 62] as well as representations [51, 81] for RL. Advances in sequence modeling led to improvements in BC [14, 11, 53, 76], RL [63, 32] and planning [42]. More recently, diffusion models [80, 38, 78] and their variants [79, 57, 5, 6] have become the popular models for large scale BC pre-training [15, 56, 89, 7], RL finetuning [73, 37], offline RL [2, 68, 87] and planning [41, 92, 58].

**NFs in RL.** In RL, NFs have been used to learn policies from uncurated data [77] and model exact MaxEnt policies [12, 33, 12]. But despite their flexibility (see Fig. 1 and Table 1), they have not been go-to models in RL; research exploring the application of NFs in RL remains limited. We believe this is because historically, NFs were thought to have limited expressivity [91, 65]. In our experiments, we show that this is not the case (Section 6). We hope that our work brings greater attention towards the potential of NFs in RL, and the development of scalable probabilistic models with similar desirable properties (Fig. 1).

## 3 Preliminaries

This section describes two major approaches for training probabilistic models and how NFs can be trained using both of them. In the next section, we show that this flexibility allows NFs to be directly plugged into various RL algorithms, improving their expressivity and performance.

**Maximum likelihood estimation (MLE).** Let $x \in \mathbb{R}^d$ be a random variable with an unknown density $p(x)$. Given a dataset of samples from this density $\mathcal{D} = \{x_i\}_{i=1}^N \sim p(x)$, the goal of MLE is to estimate the parameters of a probabilistic model $p_\theta(x)$ that maximize the likelihood of the dataset:

$$\theta_{\text{MLE}} = \arg\max_\theta \mathbb{E}_{x \sim \mathcal{D}} \big[ \log p_\theta(x) \big]. \tag{1}$$

If $p_\theta(x)$ belongs to a universal family of distributions, then the MLE estimate $p_{\theta_{\text{MLE}}}(x)$ converges towards the true density, as dataset size ($N$) increases [43].

**Variational inference (VI).** In many problems, instead of samples from the true density $p(x)$, one has access to its unnormalized part $\tilde{p}(x) = p(x) \times Z$. The goal of variational inference is to approximate $p(x)$ using a distribution $q$ from some variational family of distributions $\mathcal{Q}$:

$$q^*(x) = \arg\min_{q \in \mathcal{Q}} D_{\text{KL}}(q \,||\, p) = \arg\max_{q \in \mathcal{Q}} \mathbb{E}_{x \sim q(x)} \big[ \log \tilde{p}(x) - \log q(x) \big], \tag{2}$$

where Z can be ignored from the last equation. If $\mathcal{Q}$ is universal, the global optima of Eq. 2 will be the true density $q^*(x) = p(x)$. To optimize Eq. 2 using gradient descent, one needs to choose a family $\mathcal{Q}$, from which it is easy generate samples (the expectation requires sampling from $q$), evaluate likelihoods (second term), and backpropagate through both these operations. Due to these constraints, one typically has to restrict $\mathcal{Q}$ to be a simpler family [44] (e.g., exponential family), or resort to other inference algorithms [31] that require additional complexity (more hyper-parameters or models).

**Normalizing flows (NFs).** NFs learn an invertible mapping $f_\theta : \mathbb{R}^d \to \mathbb{R}^d$, from true density $p(x)$ to the a simple prior density $p_0(z)$. This function is parameterized such that the inverse $f_\theta^{-1}$ as well as the determinant of its jacobian matrix ($|\frac{df_\theta(x)}{dx}|$) is easily calculable. The learned density $p_\theta(x)$ can then be estimated using the change of variable formula:

$$p_\theta(x) = p_0(f_\theta(x)) |\frac{df_\theta(x)}{dx}|. \tag{3}$$

To sample from an NF, one first samples from the prior, and return the inverse function:

$$z \sim p_0(z), \text{ and return } x = f_\theta^{-1}(z). \tag{4}$$

Because Eq. (3) and Eq. (4) are passes through a single neural network, back-propagating gradients through them is straightforward. Hence, the NF can be trained efficiently using both MLE (Eq. (1)) and VI (Eq. (2)) [83, 74]:

$$\theta_{\text{MLE}} = \arg\max_\theta \mathbb{E}_{x \sim \mathcal{D}} \left[ \log p_0(f_\theta(x)) + \log(|\frac{df_\theta(x)}{dx}|) \right]. \tag{5}$$

$$\theta_{\text{VI}} = \arg\max_\theta \mathbb{E}_{\substack{z \sim p_0(z) \\ x = f_\theta^{-1}(z)}} \left[ \log \tilde{p}(x) - \log p_0(z) + \log(|\frac{df_\theta^{-1}(x)}{dx}|) \right]. \tag{6}$$

Equation (5) follows from substituting the change of variable formula in Eq. (1). Equation (6) is derived by the same substitution, followed by noting that $z = f_\theta(f_\theta^{-1}(z))$ and $-\log(|\frac{df_\theta^{-1}(x)}{dx}|) = \log(|\frac{df_\theta(x)}{dx}|)$. It is standard practice to stack multiple such mappings together $f_\theta = f_\theta^1 \circ f_\theta^1 \circ \cdots \circ f_\theta^T$. Each mapping is referred to as a *block* in the entire NF.

# 4    Many RL Algorithms do MLE and VI.

In this section, we first introduce multiple RL problems settings. We will then show how popular representative algorithms in each setting are instantiations of MLE and VI. This will (1) underscore the importance of flexible models that can be directly optimized using MLE and VI, and (2) motivate our use of NFs as a generally capable family of models for RL. We will then describe our architecture in the next section (Section 5). In all the following problems, we assume an underlying controlled markov decision process (MDP, $\mathcal{M}$), with states $s \in \mathcal{S}$ and actions $a \in \mathcal{A}$. The dynamics are $p(s'|s, a)$, the initial state distribution is $p_0(s_0)$ and $H$ is the episode horizon.

## 4.1    Offline Imitation Learning (IL)

Given a dataset of state-action trajectories $\mathcal{D} = \{s_0^i, a_0^i, \ldots, s_H^i\}_{i=1}^N$ collected by an expert(s), the aim in IL is to match the data generating behavior without any environment interactions.

**Behavior Cloning (BC) [4, 71]**    is an algorithm that models the conditional distribution of actions given states directly using MLE:

$$\arg\max_\theta \underbrace{\mathbb{E}_{s_t, a_t \sim \mathcal{D}} \left[ \log \pi_\theta(a_t \mid s_t) \right]}_{\text{MLE}}. \tag{7}$$

**Goal Conditioned Behavior Cloning (GCBC) [22, 18, 30]**    is a simple variant of BC which uses a goal $g \in \mathcal{S}$ as additional conditioning information. It is based on the insight that a trajectory leading to any goal, can be treated as an optimal trajectory to reach that goal. Given a state-action pair, a state sampled from the future $p_{t+}(s_{t+} \mid s_t, a_t)$ is treated as the goal. Formally, GCBC maximizes the following objective:

$$\arg\max_\theta \underbrace{\mathbb{E}_{\substack{s_t, a_t \sim \mathcal{D} \\ g_{t+} \sim p(s_{t+}|s_t, a_t)}} \left[ \log \pi_\theta(a_t \mid s_t, g_{t+}) \right]}_{\text{MLE}}. \tag{8}$$

Following prior work [67, 24], we set the future time $t+$ to be a truncated geometric distribution (from $t + 1$ to $H$).

## 4.2    Offline RL

Offline RL assumes an MDP $\mathcal{M}$ with a reward function $r(s, a)$. Given a dataset of trajectories $\mathcal{D} = \{s_0^i, a_0^i, r_0^i, \ldots, s_H^i, r_H^i\}_{i=1}^N$, the goal is to find a policy $\pi_\theta(a \mid s)$, which maximizes the expected discounted return $\mathbb{E}_{\pi_\theta, p}[\sum_{t=0}^H \gamma^t r(s_t, a_t)]$ [55, 49].

**MaxEnt RL + BC [28, 54, 34].** We write out a minimal objective for an offline actor-critic algorithm [28]:

$$\arg\min_{Q_\phi} \mathbb{E}_{\substack{s_t,a_t,r_t,s_{t+1}\sim D \\ a_{t+1}\sim\pi_\theta(\cdot|s_{t+1})}} \left[ \left( Q_\phi(s_t, a_t) - r_t - \gamma Q_{\bar\phi}(s_{t+1}, a_{t+1}) \right)^2 \right], \tag{9}$$

$$\arg\max_{\pi_\theta} \underbrace{\mathbb{E}_{s_t\sim D, a_t^\pi\sim\pi_\theta(\cdot|s_t)}\left[ Q_\phi(s_t, a_t^\pi) - \lambda \log \pi_\theta(a_t^\pi \mid s_t) \right]}_{\text{VI}} + \underbrace{\alpha \mathbb{E}_{s_t,a_t\sim D}\left[ \log \pi_\theta(a_t \mid s_t) \right]}_{\text{MLE}}, \tag{10}$$

where $Q_\phi(s, a)$ is the action-value function and $Q_{\bar\phi}(s, a)$ denotes the target network [61], $\lambda$ and $\alpha$ control the strength of the entropy and behavior regularization respectively. For $\lambda = 0$, this algorithm exactly matches TD3+BC [28]. In all our experiments, we use $\lambda = 0$ for minimality[2]. The policy has to sample actions that have high values while staying close to the (potentially multi-modal) behavior policy. NFs ability to be trained using both MLE and VI allows the use of expressive policies, without *any changes* to the simplest offline RL algorithms.

## 4.3 Goal conditioned RL (GCRL)

GCRL [45] is a multi-task RL problem where tasks correspond to reaching goals states $g \in \mathcal{S}$ sampled from a test goal distribution $p_{\text{test}}(g)$. We define the discounted state occupancy distribution as $p_{t+}^\pi(s_{t+} \mid s, a) = (1 - \gamma)\sum_{t=0}^\infty \gamma^t p(s_t = s_{t+} \mid s_0, a_0 = s, a)$.[3]

**Q-function estimation.** Prior work has shown that the Q-function for GCRL is equal to the discounted state occupancy distribution $Q^\pi(s, a, g) = p_{t+}^\pi(s_{t+} \mid s, a)$ [25]. Hence estimating the Q-function can be framed as a maximum likelihood problem:

$$\arg\max_\theta \underbrace{\mathbb{E}_{\substack{s_t,a_t\sim\mathcal{D} \\ g_{t+}\sim p_{t+}(s_{t+}|s_t,a_t)}}\left[ \log p_\theta(g_{t+} \mid s_t, a_t) \right]}_{\text{MLE}}. \tag{11}$$

**Unsupervised goal sampling (UGS) [26, 70]** algorithms do not assume that the test time goal distribution $p_{\text{test}}(g)$ is known a priori. The main aim of UGS is to maximize exploration by commanding goals that are neither too easy nor too difficult for the current agent. This inclines the agent to explore goals on the edge of its goal coverage. But to recognize which goals are on the edge, one needs an estimate of the goal coverage density $p_{t+}(g)$. Many methods [72, 70, 26] primarily fit a density model and then choose goals that have low density. A natural way of estimating densities is MLE, though some prior works use secondary objectives [64, 82, 26, 72] or non-parametric models [70]:

$$\arg\max_\theta \underbrace{\mathbb{E}_{\substack{s_t,a_t\sim\mathcal{D} \\ g_{t+}\sim p_{t+}(s_{t+}|s_t,a_t)}}\left[ \log p_\theta(g_{t+}) \right]}_{\text{MLE}}. \tag{12}$$

Using an estimate of this density, the canonical UGS algorithm is to sample a large batch of goals from the replay buffer and command the agent to reach the goal with the minimum density under $p_\theta(g_{t+})$ [70].

## 5 A Simple and Efficient NF Architecture for RL

This section will introduce the NF architecture we use and Section 6 will show how the this architecture fits seamlessly into many RL algorithms (those from Section 4). Many design choices affect model expressivity, the speed of computing the forward ($f_\theta(x)$) and inverse ($f_\theta^{-1}(x)$), and the efficiency of computing the determinant of the Jacobian [65]. Our desiderata is to build a *simple* and *fast* architecture that is *expressive enough* to compete and even surpass the most popular approaches across a range of RL problems. To achieve this, we build each NF block $f_\theta^t$ by leveraging two components from prior work: *(1)* a fully connected "coupling" network [20], and *(2)* a linear flow [46]. In all of our experiments, the model learns a conditional distribution $p_\theta(x \mid y)$, so each NF block $f_\theta^t : \mathbb{R}^d \times \mathbb{R}^k \to \mathbb{R}^d$ takes in $y \in \mathbb{R}^k$ as an additional input.

---

[2]This is a special case of variational inference. In bayesian inference this corresponds to approximating a maximum a-posteriori estimate.

[3]We will work with in infinite horizon case, with $H = \infty$.

**Coupling network.** We use an expressive and simple transformation proposed in Dinh et al. [19, 20]. Let $x_t$ be the input for the $t^{\text{th}}$ block and $y$ be the conditioning information. For the $t^{\text{th}}$ block, the coupling network $f_\theta^t = \{a_\theta^t, s_\theta^t : \mathbb{R}^{\lfloor d/2 \rfloor} \times \mathbb{R}^k \to \mathbb{R}^{\lceil d/2 \rceil}\}$ does the following computation:

$$(1)\ x_1^t, x_2^t = \text{split}(x^t), \quad (2)\ \tilde{x}_2^t = (x_2^t + a_\theta^t(x_1^t, y)) \times \exp(-s_\theta^t(x_1^t, y)), \quad (3)\ \tilde{x}^t = \text{concat}(x_1^t, \tilde{x}_2^t).$$

Note that the log-det Jacobian of $f_\theta^t$ is $-\text{sum}(s_\theta^t(x_1^t, y))$. Further, given $\tilde{x}^t$ and $y$, the inverse $f_\theta^{t-1}$ can be calculated as easily as the forward map (see Appendix B for details).

**Linear flow.** We adapt the generalized permutation proposed by Kingma and Dhariwal [46] to one dimensional inputs, to linearly transform the output of the coupling layer $\tilde{x}^t$. The intuition behind adding this linear flow is that each dimension of $\tilde{x}^t$ can be transformed by other dimensions in an end to end manner:

$$x^{t+1} = (\tilde{x}^t)^T W, \ \text{where } W = PL_\theta U_\theta, \ \text{and } \log|W| = \text{sum} \log(|\text{diag}(U)|).$$

Here, $P$ is a fixed permutation matrix and $L_\theta$ is lower triangular with ones on the diagonal and $U_\theta$ is upper triangular. Hence, inverting them incurs similar costs as matrix multiplication [65] and calculating the log-det Jacobian requires only the diagonal elements of $U$.

**Normalization.** While previous approaches used special variants of normalization layers ([20], [46]), we find that just using LayerNorm [3] in the coupling network's weights was sufficient of training deep NFs.

Overall, our architecture can be thought of as RealNVP [20] with generalized permutation, or a linear version of GLOW [46] without ActNorm. We have released a simple code for this architecture both in Jax and PyTorch.

# 6 Experiments

The main aim of our experiments is to use NFs and question some of the standard assumptions across several RL settings. (1) In IL, methods like diffusion policies [15] are popular because they provide expressivity but at a high computational cost. Our experiments with NFs show that this extra compute might not be necessary (Section 6.1). (2) In conditional IL, prior work argues that learning a value function is necessary for strong performance [50, 13, 29, 1]. Our ex-

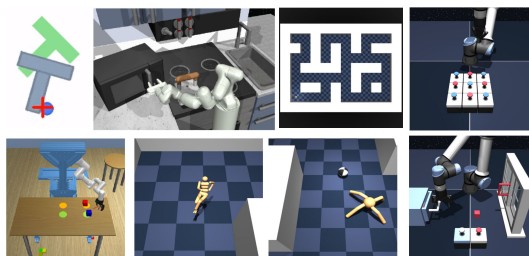

Figure 2: **Visualization of tasks**

periments suggest that a sufficiently expressive policy, such as an NF, can often suffice (Section 6.2). (3) In offline RL, auxiliary techniques are typically used to support expressive policies. We show that NF policies can be directly integrated with existing offline RL algorithms to achieve high performance (Section 6.3). (4) In UGS, we show that a single NF can estimate both the value function and goal density, enabling better exploration compared to baseline that make stronger assumptions.

In all experiments, we use the architecture described in Section 5 to implement all algorithms described in Section 4. Implementation details for all algorithms are provided in Appendix A. *Across all problems (IL, offline RL, online GCRL), we present results on a total of 82 tasks (see Fig. 2 for visualization of tasks) with 5 seeds each, which show that NFs are generally capable models that can be effectively applied to a diverse set of RL problems.* Unless stated otherwise, all error bars represent $\pm$ standard deviation across seeds. Finally, in Appendix D, we provide ablation experiments for various design choices and hyper-paramter values for some of the algorithms presented in this paper.

## 6.1 Imitation Learning (IL)

In this section, we use our NF architecture with BC (Section 4.1, Eq. (7)) for imitation learning (NF-BC). Please see implementation details of training and sampling with IL algorithm in Appendix A.1.

**Tasks.** PushT requires fine-grain control to maneuver a T-block in a fixed location. Performance is measured by the final normalized area of intersection between the block and the goal. Multimodal

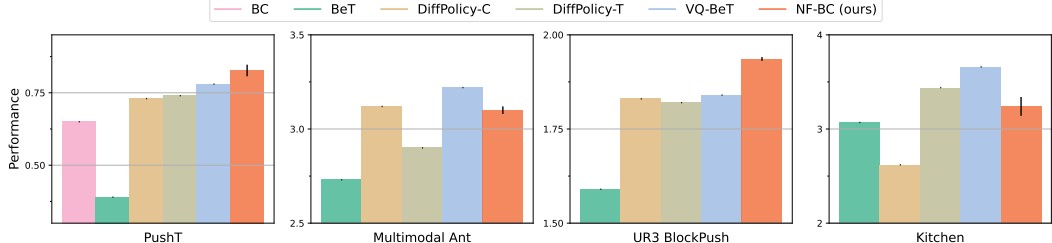

(a) **Performance comparison.** Performance is measured using normalized rewards.

| Method | Time (sec) inference; training | Params (M) | Model types | Objectives |
|---|---|---|---|---|
| NF-GCBC | 0.127 ; 0.023 | 10 M | Fully connected NFs | 1; end to end |
| VQ-BET | 0.273 ; 0.021 | 4.23 M | Transformer + VQ-VAEs | 2; req. pre-train |
| Diffusion Policy | 1.28 ; 0.019 | 65.78 M | Convolutional network | 1; end to end |

(b) **Conceptual comparison** to provide a more holistic view for RL practitioners.

Figure 3: **Imitation Learning.** Results on multi-modal behavior datasets which require fine grained control and expressivity. NF-BC is faster in inference, easier to implement (end to end optimization of a single loss function, contains 2-3 times less hyper-parameters ) and achieves competitive performance across all tasks (best success rate on two tasks and in top 2 methods on the other two tasks). In Fig. 3a, we only report standard errors for our method as baseline results were taken from prior work which did not include error bars [53]

Ant, UR3 BlockPush, and Kitchen contain multiple behaviors (4, 2, and 4 respectively). Performance is measured by the normalized number of distinct behaviors the agent is able to imitate successfully over 50 independent trials. We also report the simplicity of algorithms, as it is often a decisive factor in determining whether the algorithm can be practically employed. Simplicity is measured by the negating the number of hyperparameters used. We list and briefly describe all the hyperparameters of all the methods used to make Fig. 3a in Appendix E. **Baselines.** We compare our algorithms with 4 baselines. Diffusion policies [15] use high performing diffusion model architectures. We use both the convolutional and transformer (DiffPolicy-C and DiffPolicy-T) variant of the diffusion policy. VQ-BeT [53] uses VQ-VAEs [85] for discretizing actions and learns a transformer model over chunked representations. BeT [76] is similar to VQ-BeT but uses K-means for action chunking.

**Results.** In Fig. 3a we see that NF-BC outperforms all baselines on PushT and Kitchen and performs competitively in other two tasks. Of all the baselines, NF-BC is fastest in inference and easier to implement (Fig. 3b). For example, DiffPolicy-T uses 3 times more hyperparameters than NF-BC. These results suggest that NF policies do not lack in expressivity, compared to both diffusion and auto-regressive policies despite introducing fewer additional hyper-parameters. In fact, NF-BC can achieve these results using just a fully connected network and fraction of parameters compared to diffusion policies (see rightmost plot in Fig. 3a). Unlike BeT [76] and VQ-BeT [53], NF-BC does not use autoregressive models or VQ-VAEs (see Fig. 3b for a comparison of architectures). Reducing the number of hyperparameters is especially important for practitioners because it makes implementation easier and decreases the need for hyperparameter tuning.

## 6.2 Conditional Imitation Learning

In this section, we use our NF architecture with GCBC (Section 4.1, Eq. (8)) for conditional imitation learning (NF-GCBC).

**Tasks.** We use a total of 45 tasks from OGBench [67] meant to test diverse capabilities like learning from suboptimal trajectories with high dimensional actions, trajectory stitching and long-horizon reasoning. Performance is measured using the average success rate of reaching a goal in 50 independent trials. **Baselines.** We compare with 6 different baselines. GCBC is the GCBC 4.1 algorithm with a gaussian policy, and FM-GCBC is the GCBC 4.1 algorithm with a flow-matching based policy [57]. The velocity field for FM-GCBC is a fully connected network trained using flow

matching. FM-GCBC is the closest baseline to NF-GCBC as it uses the same underlying algorithm, uses a fully connected network like NF-GCBC, and uses an expressive family of policy models. GCIQL [48] is an offline RL algorithm that learns the optimal value function corresponding to the best actions in the dataset. QRL [88] uses a quasimetric architecture to learn the optimal value function and CRL [25] uses contrastive learning to estimate behavior policy's value function.

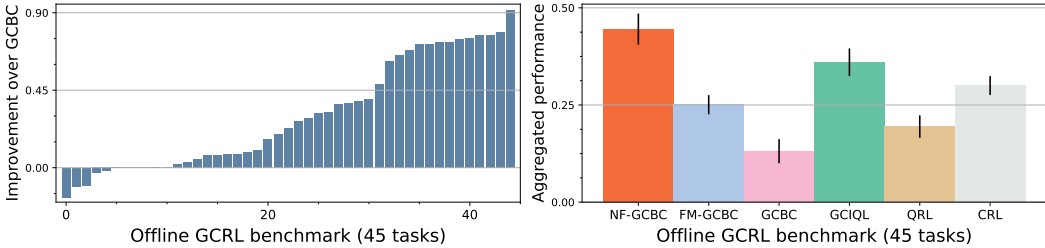

Figure 4: **Conditional Imitation Learning.** *(Left)* NF-GCBC exceeds its gaussian counterpart GCBC on $40/45$ tasks, demonstrates that using expressive models like NFs can significantly improve performance. *(Right)* Despite being a BC–based algorithm, NF-GCBC outperforms all offline RL baselines across 45 tasks. NF-GCBC performs 77% better than FM-GCBC which is the closest baseline that uses a flow matching policy with GCBC.

**Results.** In Fig. 4 (left), we see that NF-GCBC exceeds the performance (often significantly) of GCBC on $40/45$ tasks. This underscores the importance of using an expressive policy with the GCBC algorithm. In Fig. 4 (right), NF-GCBC, despite being a BC based algorithm, outperforms all baselines, including dedicated offline RL algorithms. We have added separate task wise plots in Appendix C. NF-GCBC outperforms FM-GCBC, which is the closest baseline, by 77%. While this does not necessarily imply that NFs are more expressive than flow-matching models, we hypothesize that all else being equal (e.g., fully connected architectures and identical underlying algorithms), NFs are better able to imitate the data using simple architectures as they directly optimize for log likelihoods. Flow matching models on the other hand estimate probabilities indirectly (by learning an SDE / ODE), and might require more complex architectures (transformers or convolutional networks) to work well.

### 6.3 Offline RL

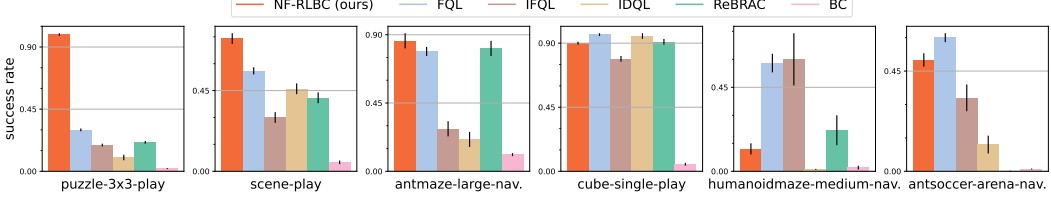

Figure 5: **Offline RL.** NF-RLBC outperforms all baselines on 15/30 tasks, and achieves top 2 results on 10/30 tasks. Notably, NF-RLBC achieves much higher results than all baselines on puzzle-3x3-play and scene-play, which require the most long horizon and sequential reasoning. For example, on puzzle-3x3-play NF-RLBC performs 230% better than the next best baseline.

While NF-GCBC performs better than offline RL algorithms on average, there are certain tasks like puzzle-3x3-play where only TD based methods succeed (Appendix C). This task, and many others as we will see, require learning a value function and accurately extracting behaviors from it. Fortunately, NFs can be trained using both VI and MLE (3). This allows us to train MaxEnt RL+BC (Section 4.2, Eq. (9)) using our NF architecture as a policy (NF-RLBC).

**Tasks.** We use a total of 30 tasks from prior work [68] testing offline RL with expressive policies. Each task has a goal which requires completing $K$ subtasks. At each step, the agent gets a reward equal to the negative of the number of subtasks left to be completed. For our experiments, we choose tasks that cover a diverse set of challenges and robots. Tasks like humanoidmaze-medium-navigate and antmaze-medium-navigate contain diverse suboptimal trajectories and high dimensional actions. Antsoccer-arena-navigate requires controlling a quadruped agent to dribble a ball to a goal location. Scene-play requires manipulating multiple objects and puzzle-3x3-play requires

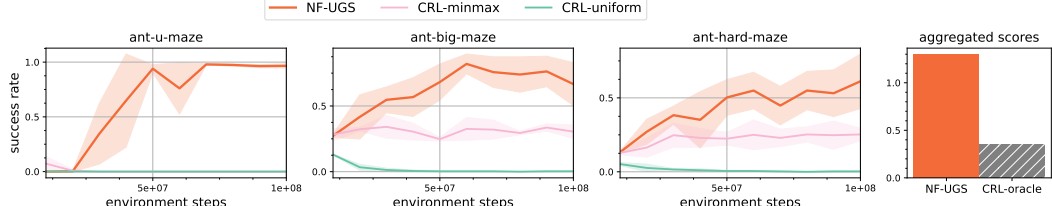

Figure 6: **GCRL.** (first 3 from left) NF-UGS consistently outperforms other unsupervised baselines. (rightmost) NF-UGS achieves higher asymptotic success rates than the CRL-oracle averaged across all tasks. Note that CRL-oracle makes extra assumptions about the availability of test-time goals.

combinatorial generalization and long-horizon reasoning. We hypothesize that the Q function for the some of these tasks (esp manipulation tasks like puzzle and scene which require long horizon reasoning) can have narrow modes of good actions and a large number of bad actions. Hence using direct gradient based optimization can be crucial to search for good actions. **Baselines.** We choose 5 representative baselines. BC performs BC with a gaussian policy, ReBRAC [84] is an offline RL algorithm with a gaussian policy that achieves impressive results on prior benchmarks [27]. IDQL [37] uses importance resampling to improve over a diffusion policy. IFQL is the flow counterpart of IDQL [68]. FQL [68] is a strong offline RL algorithm that distills a flow-matching policy into a one-step policy, while jointly maximizing the Q-function. Notably, all offline RL algorithms (IDQL, IFQL, FQL) have to incorporate additional components—such as distillation or resampling—to leverage policy expressiveness.

**Results.** Figure 5, shows that on average NF-RLBC outperforms all baselines on $15/30$ tasks, and achieves top 2 results on $10/30$ tasks. Notably, NF-RLBC achieves much higher results than all baselines on puzzle-3x3-play and scene-play, which require most long horizon and sequential reasoning (on puzzle-3x3-play NF-RLBC performs 230% better than the next best baseline). NF-RLBC is able to search for good actions because it directly uses gradient based optimization to backpropagate action gradients through the policy. All other offline RL algorithms with expressive policy classes (IDQL, IFQL, FQL) have to use extra intermediate steps.

## 6.4 Goal Conditioned RL

In this section, we use our NF architecture to estimate both the Q-function (Eq. (11)) and the coverage density (Eq. (12)). Both quantities model the distribution over the same random variable (future goals), with the Q-function conditioned on a state-action pair. We use a single neural network to estimate both by employing a masking scheme to indicate the presence or absence of conditioning [39]. We then train a goal-conditioned policy to maximize the Q-function in a standard actor-critic setup, and sample training goals using the canonical UGS algorithm (Section 4.3), resulting in our NF-UGS method.

**Tasks.** We use three problems from the UGS literature [70, 9] ant-u-maze, ant-big-maze and ant-hard-maze. These environments are difficult because the goal test distribution is unknown, and success requires thorough exploration of the entire maze. **Baselines.** We compare against three baselines. CRL-oracle [25, 9] uses contrastive learning to estimate the Q-function and serves as an oracle baseline as it uses test goals $p_{\text{test}}(g)$ to collect data during training. CRL-uniform is a variant that samples goals uniformly from the replay buffer, while CRL-minmax selects goals with the lowest Q-value from the initial state of the MDP – intuitively targeting the goals the agent believes are hardest to reach. The aim of these comparisons isn't to propose a state of the art method, but rather to check whether NFs can efficiently estimate the coverage densities of the non-stationary RL data for unsupervised exploration. **Results.** These experiments aim to evaluate whether NFs can efficiently estimate both the Q-function and the coverage density. As shown in Fig. 6, NF-UGS outperforms all baselines, and notably the oracle algorithm as well (aggregated score is the average of the final success rates across all three tasks). Unlike the minmax strategy, which plateaus over time, NF-UGS continues to improve by sampling goals to maximize the entropy of the coverage distribution [70].

# 7 Conclusion

In this paper, we view RL problems from a probabilistic perspective, asking: which probabilistic models provide the capabilities necessary for efficient RL? We argue that NFs offer a compelling answer. NFs outperform strong baselines across diverse RL settings, often with fewer parameters and lower compute, leading to simpler algorithms. By making density estimation simpler and effective, we hope our work opens new avenues for areas such as distributional RL, bayesian RL, and RL safety.

A primary limitation of NFs is that they have restrictive architectures, in that they have to be invertible. Nevertheless, there is a rich line of work characterizing which NF architectures are universal [21, 17, 65]. Another limitation of our work is that we do not explore new NF architectures; the design we use is mainly adapted from prior work (Section 5).

## Acknowledgments and Disclosure of Funding

The authors are pleased to acknowledge that the work reported on in this paper was substantially performed using the Princeton Research Computing resources at Princeton University which is consortium of groups led by the Princeton Institute for Computational Science and Engineering (PIC-SciE) and Office of Information Technology's Research Computing. The authors thank Emmanuel Bengio, Liv d'Aliberti and Eva Yi Xie for their detailed reviews on our initial drafts. The authors thank Seohong Park for providing code for some baselines and Catherine Ji for helping find a bug in our code. The authors also thank Shuangfei Zhai and the members of Princeton RL lab for helpful discussions.

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

# A Implementation details.

In this section we provide all the implementation details as well as hyper-parameters used for all the algorithms in our experiments – NF-BC, NF-GCBC, NF-RLBC, and NF-UGS. Unless specified otherwise, we chose common hyper-parameters used in prior work without tuning.

## A.1 Imitation Learning: NF-BC

For all imitation learning, we use a policy with 12 NF blocks. The fully connected networks of the coupling network for each NF block consist of two hidden layers with 512 activations each. We use a fully connected network for the state-goal encoder with 4 hidden layers having 512 activations each.

It is common in the NF literature to add some noise to the training inputs before training an NF using MLE [74, 91]. For NF-BC, we add gaussian noise to the actions before training. Hence, NF-BC models the noisy behavior policy instead $\pi^{\text{a+noise}}(a \mid s)$. To reduce noise from the obtained samples, Zhai et al. [91] propose a denoising trick:

$$z \sim p_0, \ y := f^{-1}(z) \,, x := y + \sigma^2 \nabla_y \log p_\theta(x)$$

We use this trick in all our NF-BC experiments. Table 2 contains the values of hyperparameters used by NF-BC. All experiments were done on RTX 3090s, A5000s or A6000s and did not require more than 20 hours. Experiment speed is mostly bottlenecked by intermediate evaluation of BC policies because the environments are slow and the number of rollouts per evaluation is 50. For example, the experiments on the multimodal-ant environment took 20 hours, but on pusht they were finished in less than 90 minutes.

Table 2: **NF-BC:** Hyper-parameters and their values.

| Hyper-parameter | Description |
|---|---|
| channels | 512 |
| noise std | 0.1 |
| blocks | 12 |
| encoder num layers | 4 |
| rep dims | 512 |

## A.2 Conditional Imitation Learning: NF-GCBC

For all conditional imitation learning experiments, we use a policy with 6 NF blocks. The fully connected networks of the coupling network for each NF block consist of two hidden layers with 256 activations each. We use a fully connected network for the state-goal encoder with 4 hidden layers having 512 activations each. The output representation dimensions are 64. We also employ the denoising trick proposed by [91] and mentioned in Appendix A.1. Table 3 contains the values of hyperparamters used by NF-GCBC. All experiments were done on RTX 3090s, A5000s or A6000s and did not require more than 3 hours each.

Table 3: **NF-GCBC:** Hyper-parameters and their values.

| Hyper-parameter | Description |
|---|---|
| channels | 512 |
| noise std | 0.1 |
| blocks | 6 |
| encoder num layers | 4 |
| rep dims | 512 |
| Future goal sampling discount | Table 4 |

Table 4: **NF-GCBC:** Environment values for future goal sampling discount values. For all GCBC algorithms, including the GCBC baselines, we tuned this value in the set {0.97, 0.99}

| Environment Name | Future goals sampling discount value $\gamma$ |
|---|---|
| puzzle $3 \times 3$ play | 0.97 |
| scene play | 0.97 |
| cube single play | 0.97 |
| cube double play | 0.99 |
| humanoidmaze medium navigation | 0.97 |
| antmaze large navigation | 0.99 |
| antsoccer arena navigation | 0.97 |
| antmaze medium stitch | 0.97 |
| antsoccer arena stitch | 0.97 |

## A.3 Offline RL: NF-RLBC

For all offline RL experiments, we use a policy with 6 NF blocks. The fully connected networks of the coupling network for each NF block consist of two hidden layers with 256 activations each. We use a fully connected network for the state-goal encoder with 4 hidden layers having 512 activations each. The output representation dimensions are 64. Table 5 contains the values of main hyperparameters used by NF-RLBC. All experiments were done on RTX 3090s, A5000s or A6000s and did not require more than 3 hours each.

Table 5: **NF-RLBC:** Hyper-parameters and their values.

| Hyper-parameter | Description |
|---|---|
| channels | 512 |
| noise std | 0.1 |
| actor entropy coefficient ($\lambda$ in Eq. (9)) | 0.0 |
| actor BC loss coefficient ($\alpha$ in Eq. (9)) | Table 6 |
| blocks | 6 |
| encoder num layers | 4 |
| rep dims | 512 |

Table 6: **NF-RLBC:** Environment wise actor BC loss coefficient. We use the same hyperparameter tuning procedure as all the baselines [68]. See App E.2. of Park et al. [68]

| Environment Name | Actor BC loss coefficient $\alpha$ |
|---|---|
| puzzle $3 \times 3$ play | 10.0 |
| scene play | 10.0 |
| antmaze large navigation | 1.0 |
| cube single play | 10.0 |
| humanoidmaze medium navigation | 1.0 |
| antsoccer arena navigation | 1.0 |

## A.4 Unsupervised RL: NF-UGS

For all unsupervised RL experiments, we use a goal conditioned value function with 6 NF blocks. The fully connected networks of the coupling network for each NF block consist of two hidden layers with 256 activations each. We use a fully connected network for the state-action encoder with 4 hidden layers having 1024 activations each. We train both the conditional $p_\theta(g \mid s, a)$ and the unconditional $p_\theta(g)$ using the same network. We apply a mask on the conditional variable inputs with probability 0.1. The discount factor we use is 0.99. We add a gaussian noise with standard deviation 0.05. Because we do not sample from the model in NF-UGS, we do not need to employ the denoising trick. Instead, 1024 random goals from the buffer are sampled and the one with the lowest

marginal density $p_\theta(g)$ is selected for data collection. All experiments were done on RTX 3090s, A5000s or A6000s and did not require more than 20 hours each.

# B  More details about NF block.

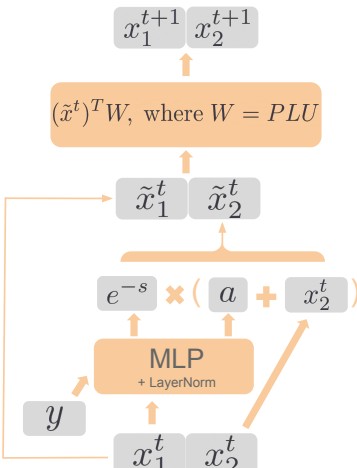

Figure 7: **NF block.**

It requires the same cost as the coupling of the forward block to calculate the coupling of the inverse block $f_\theta^{t-1}$:

$$
\begin{aligned}
&\tilde{x}_1^t, \tilde{x}_2^t = \text{split}(\tilde{x}^t), \\
&x_1^t = \tilde{x}_1^t, \\
&x_2^t = \tilde{x}_2^t \times \exp(s_\theta^t(x_1^t, y)) - a_\theta^t(x_1^t, y)), \\
&x^t = \text{concat}(x_1^t, x_2^t).
\end{aligned}
$$

See Fig. 7 for a visualization of each NF block.

Table 7: **Imitation Learning**: Performance with varying numbers of NF blocks on the Push-T task.

| number of NF blocks | Push-T |
|:---:|:---:|
| 6 | 0.77 |
| 14 | 0.79 |
| 24 | 0.827 |
| 28 | 0.859 |
| 32 | 0.85 |
| 40 | 0.844 |
| 48 | 0.87 |
| 54 | 0.85 |

Table 8: **Imitation Learning**: Performance for different NF coupling layer widths on the Push-T task.

| width of NF's coupling layers | Push-T |
|:---:|:---:|
| 128 | 0.79 |
| 256 | 0.82 |
| 1024 | 0.823 |

Table 9: **Imitation Learning**: Performance with varying noise levels for the denoising trick (Appendix A.1, [91]) on the Push-T task.

| noise_std | Push-T |
|-----------|--------|
| 0.5 | 0.77 |
| 0.3 | 0.79 |
| 0.1 | 0.827 |
| 0.05 | 0.84 |
| 0.03 | 0.873 |
| 0.01 | 0.83 |
| 0.0 | NaN |

Table 10: **Conditional Imitation Learning:** Ablation study for the importance of various components of NF-GCBC. These experiments show that the architectural modifications we propose are crucial for performance of NFs in the context of RL. We hope that the above experiments serve as sufficient motivation for our design choices.

| Method | antmaze-large-navigate | scene-play | cube-single-play |
|--------|------------------------|------------|------------------|
| NF-GCBC | 0.32 | 0.42 | 0.795 |
| without permutation flows | 0.26 | 0.04 | 0.0 |
| without LayerNorm | 0.29 | 0.3 | 0.0 |
| without sampling trick | 0.3 | 0.42 | 0.69 |

## C  Additional results.

Table 11 and Table 12 contain additional experimental results.

## D  Ablation experiments.

In Table 8, Table 7, Table 9 we ablate the values of the width of the coupling networks, the number of NF blocs, and the levels of noise for the denoising trick Appendix A.1 for NF-BC. These ablation experiments aim to show which hyper-parameters are important to tune and which hyper-parameters NF-BC is robust to.

In Table 10, we present experiments ablating the inclusion of LayerNorm, a generalized permutation layer, and the sampling trick in NF-GCBC.

Table 11: **Task wise performance for Fig. 4.**

| Environment | Score |
|-------------|-------|
| puzzle-3x3-play-v0 | $0.045_{\pm 0.005}$ |
| cube-single-play-v0 | $0.795_{\pm 0.109}$ |
| scene-play-v0 | $0.420_{\pm 0.008}$ |
| cube-double-play-v0 | $0.351_{\pm 0.087}$ |
| antmaze-medium-stitch-v0 | $0.566_{\pm 0.068}$ |
| humanoidmaze-medium-navigate-v0 | $0.314_{\pm 0.042}$ |
| antmaze-large-navigate-v0 | $0.320_{\pm 0.055}$ |
| antsoccer-arena-navigate-v0 | $0.621_{\pm 0.007}$ |
| antsoccer-arena-stitch-v0 | $0.643_{\pm 0.016}$ |

Table 12: **Task wise performance for Fig. 5.** Scores are mean ± std.

| Environment | Score |
|---|---|
| antmaze-large-navigate-singletask-task1-v0 | $0.764_{\pm 0.073}$ |
| antmaze-large-navigate-singletask-task2-v0 | $0.724_{\pm 0.088}$ |
| antmaze-large-navigate-singletask-task3-v0 | $0.924_{\pm 0.023}$ |
| antmaze-large-navigate-singletask-task4-v0 | $0.616_{\pm 0.314}$ |
| antmaze-large-navigate-singletask-task5-v0 | $0.608_{\pm 0.399}$ |
| antsoccer-arena-navigate-singletask-task1-v0 | $0.600_{\pm 0.122}$ |
| antsoccer-arena-navigate-singletask-task2-v0 | $0.556_{\pm 0.096}$ |
| antsoccer-arena-navigate-singletask-task3-v0 | $0.420_{\pm 0.077}$ |
| antsoccer-arena-navigate-singletask-task4-v0 | $0.480_{\pm 0.061}$ |
| antsoccer-arena-navigate-singletask-task5-v0 | $0.336_{\pm 0.062}$ |
| cube-single-play-singletask-task1-v0 | $0.440_{\pm 0.057}$ |
| cube-single-play-singletask-task2-v0 | $0.640_{\pm 0.087}$ |
| cube-single-play-singletask-task3-v0 | $0.768_{\pm 0.060}$ |
| cube-single-play-singletask-task4-v0 | $0.600_{\pm 0.082}$ |
| cube-single-play-singletask-task5-v0 | $0.616_{\pm 0.067}$ |
| humanoidmaze-medium-navigate-singletask-task1-v0 | $0.036_{\pm 0.045}$ |
| humanoidmaze-medium-navigate-singletask-task2-v0 | $0.188_{\pm 0.097}$ |
| humanoidmaze-medium-navigate-singletask-task3-v0 | $0.148_{\pm 0.072}$ |
| humanoidmaze-medium-navigate-singletask-task4-v0 | $0.016_{\pm 0.015}$ |
| humanoidmaze-medium-navigate-singletask-task5-v0 | $0.208_{\pm 0.053}$ |
| puzzle-3x3-play-singletask-task1-v0 | $0.996_{\pm 0.008}$ |
| puzzle-3x3-play-singletask-task2-v0 | $0.996_{\pm 0.008}$ |
| puzzle-3x3-play-singletask-task3-v0 | $0.976_{\pm 0.029}$ |
| puzzle-3x3-play-singletask-task4-v0 | $0.996_{\pm 0.008}$ |
| puzzle-3x3-play-singletask-task5-v0 | $0.996_{\pm 0.008}$ |
| scene-play-singletask-task1-v0 | $1.000_{\pm 0.000}$ |
| scene-play-singletask-task2-v0 | $0.880_{\pm 0.075}$ |
| scene-play-singletask-task3-v0 | $0.988_{\pm 0.016}$ |
| scene-play-singletask-task4-v0 | $0.856_{\pm 0.056}$ |
| scene-play-singletask-task5-v0 | $0.000_{\pm 0.000}$ |

# E   Hyperparameters used by behavior cloning algorithms based on different probabilistic models.

Table 13: **Hyper-parameters used by NF-BC.**

| Hyper-parameter | Description |
|---|---|
| channels | width of each layer of the coupling MLP network. |
| noise std | added noise to labels for training the NF using MLE. |
| blocks | number of NF blocks. |
| encoder num layers | number of layers in the observation encoder MLP network. |
| rep dims | output dimension of observation encoder MLP network. |

Table 14: **Hyper-parameters used by BET.**

| Hyper-parameter | Description |
|---|---|
| embedding dims | dimension length for attention vectors. |
| num layers | number of attention blocks. |
| num heads | number of heads in attention. |
| act scale | action scaling parameter. |
| num bins | number of centroids for k-means. |
| offset prediction | whether to predict offsets for actions or not. |
| offset loss scale | action offset prediction loss coefficient. |
| gamma | focal loss parameter. |

Table 15: **Hyper-parameters used by DiffPolicy-C.**

| Hyper-parameter | Description |
|---|---|
| time-step embedding dims | embedding dimension of diffusion timestep. |
| down dims | downsampling dimensions for Unet architecture. |
| kernel size | kernel size for CNNs in Unet. |
| n groups | number of groups for group normalization. |
| scheduler train timesteps | Length of the forward diffusion chain while training. |
| num inference steps | Length of the denoising steps while sampling. |
| beta schedule type | type of scheduler for noising. |
| beta start | first-step noise for diffusion scheduler. |
| beta end | last-step noise for diffusion scheduler. |
| ema inverse gamma | parameter for ema rate schedule. |
| ema power | parameter for ema rate schedule. |
| ema min | minimum value of ema decay rate. |
| ema max | maximum value of ema decay rate. |

Table 16: **Hyper-parameters used by DiffPolicy-T.**

| Hyper-parameter | Description |
|---|---|
| time-step embedding dims | embedding dimension of diffusion timestep. |
| embedding dims | dimension length for attention vectors. |
| num layers | number of attention blocks. |
| num heads | number of heads in attention. |
| scheduler train timesteps | Length of the forward diffusion chain while training. |
| num inference steps | Length of the denoising steps while sampling. |
| beta schedule type | type of scheduler for noising. |
| beta start | first-step noise for diffusion scheduler. |
| beta end | last-step noise for diffusion scheduler. |
| ema inverse gamma | parameter for ema rate schedule. |
| ema power | parameter for ema rate schedule. |
| ema min | minimum value of ema decay rate. |
| ema max | maximum value of ema decay rate. |

Table 17: **Hyper-parameters used by VQ-BET.**

| Hyper-parameter | Description |
|---|---|
| embedding dims | dimension length for attention vectors. |
| num layers | number of attention blocks. |
| num heads | number of heads in attention. |
| action seq len | sequence length used to train the behavior transformer. |
| vqvae num embed | cookbook size of vqvae. |
| vqvae embedding dims | embedding dimensions of vqvae codes. |
| vqvae groups | number of groups in residual vqvae. |
| encoder loss multiplier | encoder loss coefficient. |
| offset loss multiplier | action offset prediction loss coefficient. |
| secondary code multiplier | secondary code prediction loss coefficient. |
| gamma | focal loss parameter. |

## F   Societal Impacts

**Positive Impacts.** Our work directly improves various RL algorithms in various RL problem settings, which can benefit domains like robotics, healthcare, and energy systems by enabling efficient autonomous decision-making.

**Negative Impacts.** As with many advances in reinforcement learning, improvements in policy expressiveness can be misused in domains such as surveillance, autonomous weapons, or manipulation in recommendation systems. Further, increased training complexity may lead to higher computational costs, raising concerns about energy use and environmental impact.

## NeurIPS Paper Checklist

1. **Claims**

   Question: Do the main claims made in the abstract and introduction accurately reflect the paper's contributions and scope?

   Answer: [Yes]

   Justification: The main claims made in the abstract and introduction are that our experiments show that NFs are expressive, flexible and generally capable models for many RL problems like imitation learning, offline, goal conditioned RL and unsupervised RL. Towards the end of our introduction, we highlight the success results of NF based algorithms in all of these problem settings. Section 6 contains detailed discussions of these experiments and results which concretely justify these claims.

   Guidelines:

   - The answer NA means that the abstract and introduction do not include the claims made in the paper.
   - The abstract and/or introduction should clearly state the claims made, including the contributions made in the paper and important assumptions and limitations. A No or NA answer to this question will not be perceived well by the reviewers.
   - The claims made should match theoretical and experimental results, and reflect how much the results can be expected to generalize to other settings.
   - It is fine to include aspirational goals as motivation as long as it is clear that these goals are not attained by the paper.

2. **Limitations**

   Question: Does the paper discuss the limitations of the work performed by the authors?

   Answer: [Yes]

   Justification: Yes, the second paragraph of Section 7 discussion the main limitations of our work.

   Guidelines:

- The answer NA means that the paper has no limitation while the answer No means that the paper has limitations, but those are not discussed in the paper.
- The authors are encouraged to create a separate "Limitations" section in their paper.
- The paper should point out any strong assumptions and how robust the results are to violations of these assumptions (e.g., independence assumptions, noiseless settings, model well-specification, asymptotic approximations only holding locally). The authors should reflect on how these assumptions might be violated in practice and what the implications would be.
- The authors should reflect on the scope of the claims made, e.g., if the approach was only tested on a few datasets or with a few runs. In general, empirical results often depend on implicit assumptions, which should be articulated.
- The authors should reflect on the factors that influence the performance of the approach. For example, a facial recognition algorithm may perform poorly when image resolution is low or images are taken in low lighting. Or a speech-to-text system might not be used reliably to provide closed captions for online lectures because it fails to handle technical jargon.
- The authors should discuss the computational efficiency of the proposed algorithms and how they scale with dataset size.
- If applicable, the authors should discuss possible limitations of their approach to address problems of privacy and fairness.
- While the authors might fear that complete honesty about limitations might be used by reviewers as grounds for rejection, a worse outcome might be that reviewers discover limitations that aren't acknowledged in the paper. The authors should use their best judgment and recognize that individual actions in favor of transparency play an important role in developing norms that preserve the integrity of the community. Reviewers will be specifically instructed to not penalize honesty concerning limitations.

3. **Theory assumptions and proofs**

   Question: For each theoretical result, does the paper provide the full set of assumptions and a complete (and correct) proof?

   Answer: [NA]

   Our paper is primarily an empirical paper. While Section 4 does provide a unifying probabilistic framework to think about many RL algorithms, the specific algorithm-wise results are known and have been cited accordingly.

   Guidelines:

   - The answer NA means that the paper does not include theoretical results.
   - All the theorems, formulas, and proofs in the paper should be numbered and cross-referenced.
   - All assumptions should be clearly stated or referenced in the statement of any theorems.
   - The proofs can either appear in the main paper or the supplemental material, but if they appear in the supplemental material, the authors are encouraged to provide a short proof sketch to provide intuition.
   - Inversely, any informal proof provided in the core of the paper should be complemented by formal proofs provided in appendix or supplemental material.
   - Theorems and Lemmas that the proof relies upon should be properly referenced.

4. **Experimental result reproducibility**

   Question: Does the paper fully disclose all the information needed to reproduce the main experimental results of the paper to the extent that it affects the main claims and/or conclusions of the paper (regardless of whether the code and data are provided or not)?

   Answer: [Yes]

   Yes, the paper provide all information needed to reproduce the all the experimental results. Appendix A contains the implementation details of all the algorithms, including the hyperparameters used and their values. We have also provided the code for all our experiments organized in a readable manner in supplementary experiments. We have also provided the code for our architecture in both Jax [10] and PyTorch [69].

Guidelines:

- The answer NA means that the paper does not include experiments.
- If the paper includes experiments, a No answer to this question will not be perceived well by the reviewers: Making the paper reproducible is important, regardless of whether the code and data are provided or not.
- If the contribution is a dataset and/or model, the authors should describe the steps taken to make their results reproducible or verifiable.
- Depending on the contribution, reproducibility can be accomplished in various ways. For example, if the contribution is a novel architecture, describing the architecture fully might suffice, or if the contribution is a specific model and empirical evaluation, it may be necessary to either make it possible for others to replicate the model with the same dataset, or provide access to the model. In general. releasing code and data is often one good way to accomplish this, but reproducibility can also be provided via detailed instructions for how to replicate the results, access to a hosted model (e.g., in the case of a large language model), releasing of a model checkpoint, or other means that are appropriate to the research performed.
- While NeurIPS does not require releasing code, the conference does require all submissions to provide some reasonable avenue for reproducibility, which may depend on the nature of the contribution. For example
  - (a) If the contribution is primarily a new algorithm, the paper should make it clear how to reproduce that algorithm.
  - (b) If the contribution is primarily a new model architecture, the paper should describe the architecture clearly and fully.
  - (c) If the contribution is a new model (e.g., a large language model), then there should either be a way to access this model for reproducing the results or a way to reproduce the model (e.g., with an open-source dataset or instructions for how to construct the dataset).
  - (d) We recognize that reproducibility may be tricky in some cases, in which case authors are welcome to describe the particular way they provide for reproducibility. In the case of closed-source models, it may be that access to the model is limited in some way (e.g., to registered users), but it should be possible for other researchers to have some path to reproducing or verifying the results.

5. **Open access to data and code**

   Question: Does the paper provide open access to the data and code, with sufficient instructions to faithfully reproduce the main experimental results, as described in supplemental material?

   Answer: [Yes]

   Like mentioned in the previous answer, we have provided the code for all our experiments organized in a readable manner in supplementary experiments. We have also provided the code for our architecture in both Jax [10] and PyTorch [69].

   Guidelines:

   - The answer NA means that paper does not include experiments requiring code.
   - Please see the NeurIPS code and data submission guidelines (https://nips.cc/public/guides/CodeSubmissionPolicy) for more details.
   - While we encourage the release of code and data, we understand that this might not be possible, so "No" is an acceptable answer. Papers cannot be rejected simply for not including code, unless this is central to the contribution (e.g., for a new open-source benchmark).
   - The instructions should contain the exact command and environment needed to run to reproduce the results. See the NeurIPS code and data submission guidelines (https://nips.cc/public/guides/CodeSubmissionPolicy) for more details.
   - The authors should provide instructions on data access and preparation, including how to access the raw data, preprocessed data, intermediate data, and generated data, etc.
   - The authors should provide scripts to reproduce all experimental results for the new proposed method and baselines. If only a subset of experiments are reproducible, they should state which ones are omitted from the script and why.

- At submission time, to preserve anonymity, the authors should release anonymized versions (if applicable).
- Providing as much information as possible in supplemental material (appended to the paper) is recommended, but including URLs to data and code is permitted.

6. **Experimental setting/details**

Question: Does the paper specify all the training and test details (e.g., data splits, hyperparameters, how they were chosen, type of optimizer, etc.) necessary to understand the results?

Answer: [Yes]

Appendix A contains the implementation details of all the algorithms, including the hyperparameters used, their values, and details of how they were chosen. Every subsection of Section 6 contains detailed description of the tasks and baselines for all RL problems (offline imitation learning, conditional offline imitation learning, offline RL, unsupervised goal conditioned RL).

Guidelines:

- The answer NA means that the paper does not include experiments.
- The experimental setting should be presented in the core of the paper to a level of detail that is necessary to appreciate the results and make sense of them.
- The full details can be provided either with the code, in appendix, or as supplemental material.

7. **Experiment statistical significance**

Question: Does the paper report error bars suitably and correctly defined or other appropriate information about the statistical significance of the experiments?

Answer: [Yes]

Justification: Yes, all experiments are conducted across five seeds and error bars are shown for all bar / graph plots.

Guidelines:

- The answer NA means that the paper does not include experiments.
- The authors should answer "Yes" if the results are accompanied by error bars, confidence intervals, or statistical significance tests, at least for the experiments that support the main claims of the paper.
- The factors of variability that the error bars are capturing should be clearly stated (for example, train/test split, initialization, random drawing of some parameter, or overall run with given experimental conditions).
- The method for calculating the error bars should be explained (closed form formula, call to a library function, bootstrap, etc.)
- The assumptions made should be given (e.g., Normally distributed errors).
- It should be clear whether the error bar is the standard deviation or the standard error of the mean.
- It is OK to report 1-sigma error bars, but one should state it. The authors should preferably report a 2-sigma error bar than state that they have a 96% CI, if the hypothesis of Normality of errors is not verified.
- For asymmetric distributions, the authors should be careful not to show in tables or figures symmetric error bars that would yield results that are out of range (e.g. negative error rates).
- If error bars are reported in tables or plots, The authors should explain in the text how they were calculated and reference the corresponding figures or tables in the text.

8. **Experiments compute resources**

Question: For each experiment, does the paper provide sufficient information on the computer resources (type of compute workers, memory, time of execution) needed to reproduce the experiments?

Answer: [Yes]

Justification: In Appendix A, we provide the type of GPUs used and the time taken for all our experiments.

Guidelines:

- The answer NA means that the paper does not include experiments.
- The paper should indicate the type of compute workers CPU or GPU, internal cluster, or cloud provider, including relevant memory and storage.
- The paper should provide the amount of compute required for each of the individual experimental runs as well as estimate the total compute.
- The paper should disclose whether the full research project required more compute than the experiments reported in the paper (e.g., preliminary or failed experiments that didn't make it into the paper).

9. **Code of ethics**

   Question: Does the research conducted in the paper conform, in every respect, with the NeurIPS Code of Ethics https://neurips.cc/public/EthicsGuidelines?

   Answer: [Yes]

   Justification: I have read the guidelines and our research does indeed conform, in every respect, with the NeurIPS Code of Ethics.

   Guidelines:

   - The answer NA means that the authors have not reviewed the NeurIPS Code of Ethics.
   - If the authors answer No, they should explain the special circumstances that require a deviation from the Code of Ethics.
   - The authors should make sure to preserve anonymity (e.g., if there is a special consideration due to laws or regulations in their jurisdiction).

10. **Broader impacts**

    Question: Does the paper discuss both potential positive societal impacts and negative societal impacts of the work performed?

    Answer: [Yes]

    Justification: In Appendix F we have added a brief discussion of both positive and potentially negative societal impacts of our work.

    Guidelines:

    - The answer NA means that there is no societal impact of the work performed.
    - If the authors answer NA or No, they should explain why their work has no societal impact or why the paper does not address societal impact.
    - Examples of negative societal impacts include potential malicious or unintended uses (e.g., disinformation, generating fake profiles, surveillance), fairness considerations (e.g., deployment of technologies that could make decisions that unfairly impact specific groups), privacy considerations, and security considerations.
    - The conference expects that many papers will be foundational research and not tied to particular applications, let alone deployments. However, if there is a direct path to any negative applications, the authors should point it out. For example, it is legitimate to point out that an improvement in the quality of generative models could be used to generate deepfakes for disinformation. On the other hand, it is not needed to point out that a generic algorithm for optimizing neural networks could enable people to train models that generate Deepfakes faster.
    - The authors should consider possible harms that could arise when the technology is being used as intended and functioning correctly, harms that could arise when the technology is being used as intended but gives incorrect results, and harms following from (intentional or unintentional) misuse of the technology.
    - If there are negative societal impacts, the authors could also discuss possible mitigation strategies (e.g., gated release of models, providing defenses in addition to attacks, mechanisms for monitoring misuse, mechanisms to monitor how a system learns from feedback over time, improving the efficiency and accessibility of ML).

11. **Safeguards**

Question: Does the paper describe safeguards that have been put in place for responsible release of data or models that have a high risk for misuse (e.g., pretrained language models, image generators, or scraped datasets)?

Answer: [NA]

Justification: We do not use any of data or models that have a high risk for misuse.

Guidelines:

- The answer NA means that the paper poses no such risks.
- Released models that have a high risk for misuse or dual-use should be released with necessary safeguards to allow for controlled use of the model, for example by requiring that users adhere to usage guidelines or restrictions to access the model or implementing safety filters.
- Datasets that have been scraped from the Internet could pose safety risks. The authors should describe how they avoided releasing unsafe images.
- We recognize that providing effective safeguards is challenging, and many papers do not require this, but we encourage authors to take this into account and make a best faith effort.

12. **Licenses for existing assets**

Question: Are the creators or original owners of assets (e.g., code, data, models), used in the paper, properly credited and are the license and terms of use explicitly mentioned and properly respected?

Answer: [Yes]

Justification: Wherever necessary, we have cited the original source of the datasets and benchmarks used in our paper Section 6.

Guidelines:

- The answer NA means that the paper does not use existing assets.
- The authors should cite the original paper that produced the code package or dataset.
- The authors should state which version of the asset is used and, if possible, include a URL.
- The name of the license (e.g., CC-BY 4.0) should be included for each asset.
- For scraped data from a particular source (e.g., website), the copyright and terms of service of that source should be provided.
- If assets are released, the license, copyright information, and terms of use in the package should be provided. For popular datasets, paperswithcode.com/datasets has curated licenses for some datasets. Their licensing guide can help determine the license of a dataset.
- For existing datasets that are re-packaged, both the original license and the license of the derived asset (if it has changed) should be provided.
- If this information is not available online, the authors are encouraged to reach out to the asset's creators.

13. **New assets**

Question: Are new assets introduced in the paper well documented and is the documentation provided alongside the assets?

Answer: [Yes]

Justification: Yes, our code is well documented. Details about the architecture, experiments and results have been provided in the paper Section 6, Appendix A.

Guidelines:

- The answer NA means that the paper does not release new assets.
- Researchers should communicate the details of the dataset/code/model as part of their submissions via structured templates. This includes details about training, license, limitations, etc.

- The paper should discuss whether and how consent was obtained from people whose asset is used.
- At submission time, remember to anonymize your assets (if applicable). You can either create an anonymized URL or include an anonymized zip file.

14. **Crowdsourcing and research with human subjects**

Question: For crowdsourcing experiments and research with human subjects, does the paper include the full text of instructions given to participants and screenshots, if applicable, as well as details about compensation (if any)?

Answer: [NA]

Justification: There are no crowdsourcing experiments in our paper.

Guidelines:

- The answer NA means that the paper does not involve crowdsourcing nor research with human subjects.
- Including this information in the supplemental material is fine, but if the main contribution of the paper involves human subjects, then as much detail as possible should be included in the main paper.
- According to the NeurIPS Code of Ethics, workers involved in data collection, curation, or other labor should be paid at least the minimum wage in the country of the data collector.

15. **Institutional review board (IRB) approvals or equivalent for research with human subjects**

Question: Does the paper describe potential risks incurred by study participants, whether such risks were disclosed to the subjects, and whether Institutional Review Board (IRB) approvals (or an equivalent approval/review based on the requirements of your country or institution) were obtained?

Answer: [NA]

Justification: the paper does not involve crowdsourcing nor research with human subjects.

Guidelines:

- The answer NA means that the paper does not involve crowdsourcing nor research with human subjects.
- Depending on the country in which research is conducted, IRB approval (or equivalent) may be required for any human subjects research. If you obtained IRB approval, you should clearly state this in the paper.
- We recognize that the procedures for this may vary significantly between institutions and locations, and we expect authors to adhere to the NeurIPS Code of Ethics and the guidelines for their institution.
- For initial submissions, do not include any information that would break anonymity (if applicable), such as the institution conducting the review.

16. **Declaration of LLM usage**

Question: Does the paper describe the usage of LLMs if it is an important, original, or non-standard component of the core methods in this research? Note that if the LLM is used only for writing, editing, or formatting purposes and does not impact the core methodology, scientific rigorousness, or originality of the research, declaration is not required.

Answer: [NA]

Justification: The core method development in this research does not involve LLMs as any important, original, or non-standard components.

Guidelines:

- The answer NA means that the core method development in this research does not involve LLMs as any important, original, or non-standard components.
- Please refer to our LLM policy (https://neurips.cc/Conferences/2025/LLM) for what should or should not be described.

