# OpenReview forum: "Normalizing Flows are Capable Models for Continuous Control"
_NeurIPS.cc/2025/Conference — NeurIPS 2025 poster_

### Official Review · Reviewer_aQtu · 2025-06-14

**Clarity:** 3
**Significance:** 2
**Originality:** 2
**Rating:** 4
**Confidence:** 5

**Summary:**

This paper proposes using normalizing flows (NF) to model policy distributions (i.e., action distributions) in offline reinforcement learning (RL) and state occupancy distributions in goal-conditioned RL (GCRL). The authors demonstrate that, due to NF’s ability to support both maximum likelihood estimation (MLE) and variational inference (VI), the objectives of offline RL and GCRL can be computed efficiently. Experimental results show that the proposed approach improves performance on certain tasks in imitation learning (IL), offline RL, and GCRL.

**Questions:**

1. Given the conceptual overlap with prior work [1–6], could the authors provide experimental comparisons with these baselines? Also, including recent GCRL approaches such as [7] would improve the completeness of the evaluation.
2. Could the authors include an example to illustrate the benefits of NFs over other continuous generative models (e.g., VAEs, EBMs, and diffusion)? A minimal setup, similar to Section 5.2 in [8] or Section 4.1 in [3], could provide helpful intuition for why NFs may be preferable in offline RL and GCRL.
3. The simplicity metric in Fig. 3 seems oversimplified and biased toward counting hyperparameters. This overlooks their sensitivity and broader implementation complexity. Could the authors elaborate on the rationale behind this simplicity metric and why the sensitivity is not considered?
4. Can the authors provide performance comparisons between NF-GCBC and GCIQL in a manner similar to Fig. 5 (Left)?
5. Since NFs are inherently continuous generative models, they are not directly applicable to environments with discrete action (Sections 4.1 and 4.2) or state (Section 4.3) spaces. In these cases, the claim that “Normalizing Flows are Capable Models for RL” may be overstated. Could the authors clarify how their approach can generalize to discrete settings and provide empirical evidence?
6. The scope of this work is limited to IL, offline RL, and GCRL. Is there a specific reason online RL was not considered? Could the authors comment on the applicability of NFs to online RL?

\
**References**

[1] Haarnoja et al. Latent space policies for hierarchical reinforcement learning. ICML 2018. \
[2] Mazoure et al. Leveraging Exploration in Off-policy Algorithms via Normalizing Flows. CoRL 2019. \
[3] Chao et al. Maximum entropy reinforcement learning via energy-based normalizing flow. NeurIPS 2024. \
[4] Akimov et al. Let Offline RL Flow: Training Conservative Agents in the Latent Space of Normalizing Flows. NeurIPS Workshop 2022. \
[5] Liu et al. Uncertainty-Aware Reinforcement Learning for Risk-Sensitive Player Evaluation in Sports Game. NeurIPS 2022. \
[6] Yang et al. Flow to Control: Offine Reinforcement Learning with Lossless Primitive Discovery. AAAI 2023. \
[7] Park et al. HIQL: Offline Goal-Conditioned RL with Latent States as Actions. NeurIPS 2023. \
[8] Haarnoja et al. Reinforcement Learning with Deep Energy-Based Policies. ICML 2017.

**Ethical Concerns:**

["NO or VERY MINOR ethics concerns only"]

**Final Justification:**

My questions were addressed. I increased my score to a borderline accept.

**Limitations:**

Yes. The authors adequately addressed the limitations and potential negative societal impact.

**Paper Formatting Concerns:**

No major formatting issue.

**Quality:**

2

**Strengths And Weaknesses:**

**Strengths**
1. The proposed method is simple, with writing that is easy to follow.
2. The paper provides a thorough discussion of the loss functions and presents a straightforward implementation based on Glow and RealNVP architectures.
3. Experimental results demonstrate performance improvements when applying NF to some of the selected RL tasks.

**Weaknesses**
1. The use of NFs in both online RL [1–3] and offline RL [4–6] has been well-explored in prior work. This paper largely repurposes NFs in the context of offline RL and GCRL without introducing fundamentally new insights. (See Question 1)
2. While the paper spans a broad range of topics, it does not deeply investigate why NFs are particularly suited for offline RL and GCRL. Many generative models (e.g., VAEs, EBMs, and diffusion models) also support MLE and VI, as shown in Table 1. A more thorough justification—beyond ease of loss computation or sampling—is needed. (See Question 2)
3. The proposed approach underperforms on a number of benchmarks. For instance, in IL (Fig. 3), NF-BC is inferior to VQ-BeT on the Kitchen and Multi-modal Ant tasks, while also requiring twice as many learnable parameters. In the offline RL experiments (Fig. 6), NF-RLBC is outperformed by FQL on tasks such as `antsoccer-arena-nav`, `humanoidmaze-medium-nav`, and `cube-single-play`.
4. Minor technical inaccuracy: Line 29 claims that training diffusion models requires solving an ODE. However, standard diffusion models are trained using denoising score matching, which is simulation-free and computationally efficient.

---

> ### Author Rebuttal · Authors · 2025-07-30
>
> We thank the reviewer for their insightful feedback and constructive comments. The reviewer’s main feedback is:
>
> (1) regarding the conceptual overlap and comparisons with prior work.
>
> (2) a deeper investigation of why NFs are particularly suited for RL.
>
> (3) analysis of hyperparameter sensitivity and broader implementation complexity.
>
> For (1), we agree that there are prior papers that apply NFs to RL. Our paper is conceptually different in ways we highlight below. We have revised the related work section and introduction to better explain and clarify these important conceptual differences. Moreover, we have added new experiments comparing the architecture of SAC-LSP [1] with the architecture proposed in our paper.
>
> For (2), we have added new ablation experiments, in benchmark tasks and a didactic task. These experiments show that expressivity and unbiased VI are the primary properties of NFs which make them particularly suited for RL algorithms.
>
> For (3), we add comparisons for the number of model parameters (complexity), number of training objectives (end to end vs step wise) with baselines, and provide plots for hyperparameters sensitivity of our method to give a holistic view of implementation simplicity for practitioners.
>
> **Do these answers, revisions and experiments address all the reviewer's concerns?**
>
> > Comparison with the architecture of [1], on conditional IL and offline RL
>
> **Success rates (higher is better):**
> | Method  | scene-play | cube-single-play |
> |---------|---------|----------|
> | NF-GCBC      | 0.42 | 0.795 |
> | GCBC-LSP | 0.02 |  0.04 |
>
> | Method  | antmaze-large-navigate | antsoccer-arena-navigate | cube-single-play | puzzle-3x3-play |  scene-play | humanoidmaze-medium-navigate |
> |---------|---------|----------|---------|----------|----------| ----------|
> | SAC-NF      | 0.76 | 0.6 | 0.440 | 0.996 | 1.0 | 0.03 |
> | SAC-LSP | 0.86 | 0.43 | 0.03 | 0.26 | 0.49 | 0.0 |
>
> SAC-LSP uses the RealNVP architecture, which we find to be sometimes unstable to train (exploding likelihoods). **These results show our architecture is more stable and performant.**
>
> > conceptual overlap with [2–6]
>
> We agree that these papers use different versions of NFs in the context of RL, but other than that there are distinct conceptual / methodological differences. We first emphasize the main argument of our paper -- because NFs do MLE and VI efficiently, using them as plug and play models in various RL algorithms can reap significant performance gains, rivaling and even surpassing SOTA / complex RL algorithms. We will now note how our argument is conceptually different than the papers pointed out by the reviewer:
>
> [2] use a high level policy ( $p(z \mid s)$ ) and an NF low level policy ( $p(a \mid z)$ ). Hence they can only obtain a lower bound on the true log-likelihood of the policy $E_{p(z \mid s)} \log [ \pi(a \mid z) ] \leq \log E_{p(z \mid s)} [ \pi(a \mid z) ] = \log \pi(a \mid z) $ (see 5th line of sec 6 or sec 6.1 where they allude to this).
>
> [3] Use a special kind of NF called Energy-Based Normalizing Flows, this puts constraints on the NF (state independent part of jacobian det should be equal to the value function, and for the state independent part be equal to the Q function). These constraints don't satisfy in general and hence require new techniques like LRS or SCDQ (see sec 3.2 of [3]).
>
> [4] Like [2], they also train a high level policy followed by an NF low level policy. Moreover, they train the NF using weighted regression and introduce new techniques for conservatism (see sec 3.1). This is in contrast to our approach of doing VI directly using NFs.
>
> [5] Does not use NFs directly for RL, but only uses them for uncertainty estimation (sec 4.2). Both the architectures introduced and the tasks considered in our papers are very different.
>
> [6] The main reason they use an NF is because they require an invertible network. Moreover their algorithm is a skill learning algorithm, again optimizing an evidence lower bound. This is in contrast to our argument of doing VI directly using NFs. Additionally, a comparison between a hierarchical algorithm with our flat algorithm wouldn't be fair.
>
> **Our central insight is that NFs unify MLE and VI objectives in RL while being expressive, stable, and fast to train. This allows them to serve as plug-and-play models without needing additional priors or multi-stage training. This is in contrast to prior works that use NFs in nested or constrained forms ([2-4,6]). By showing this across diverse RL settings, we believe our work repositions NFs as a general tool in the RL toolbox, not just a modeling novelty.**
>
> > deeper investigation why NFs are particularly suited for RL
>
> The main reasons why NFs perform so well is their expressivity and unbiased VI (see the next 4 goal experiment). Almost all datasets we use have multi-modal and noisy distributions. We compare the validation likelihoods of the policies learnt by GCBC and NF-GCBC on three tasks:
>
> **Log-likelihoods (higher is better):**
> | Method | antmaze-large-navigate | scene-play | cube-single-play |
> |---------|----------|----------|----------|
> | NF -GCBC     | -0.673   | 2.975 | 2.756 |
> | GCBC | -7.662 | -4.65 |  -4.661 |
>
> NF-GCBC is able to model the behavior policies more accurately. It is well established that higher-likelihoods are correlated with performance and generalisation in imitation learning [1,2].
>
> We have also added new experiments to show the contribution of the components we use:
>
> **Success rates (higher is better):**
> | Method | antmaze-large-navigate | scene-play | cube-single-play |
> |---------|----------|----------|----------|
> | NF -GCBC     | 0.32   | 0.42 | 0.795 |
> | without permutation flows | 0.26 | 0.04 |  0.0 |
> | without LayerNorm | 0.29 | 0.3 |  0.0 |
> | without sampling trick | 0.3 | 0.42 |  0.69 |
>
> These experiments show that our architectural modifications are important for performance of NFs in the context of RL. We hope that the above experiments serve as sufficient motivation for our design choices.
>
> > illustrate the benefits of NFs over other continuous generative models (e.g., VAEs, EBMs, and diffusion)
>
> In our main paper, we compare and show benefits over SOTA algorithms using VAEs, EBMs and diffusion. Fig 3 compares with diffusion policy / VQ-BET, Fig 5 with FM-GCBC (flow matching), Fig 6 with FQL, IFQL, IDQL, all use flow / diffusion models, Fig 6 compares with CRL (contrastive energy based models). **Of all the 85 tasks, our proposed algorithms exceed or match in 65 and is amongst the top two in seven tasks**. As requested by the reviewer, we have further added a didactic 4 goal environment (sec 4.1 in [3]):
>
> **Goal reaching distribution:**
> | Method | goal 1 | goal 2 | goal 3 | goal 4 |
> |---------|----------|----------|----------|----------|
> | SAC     | 0   | 0 | 0 | 20 |
> | SAC-VAE | 0 | 0 |  0 | 20 |
> | SAC-NF | 8 | 2 |  7 | 3 |
>
> SAC-NF is able to reach all 4 goals, whereas SAC-VAE, which uses a VAE state encoder and a gaussian action decoder still shows mode-seeking behavior, a problem faced by reverse KL optimization. This defect of SAC-VAE can also be caused because only a lower bound on the policy entropy is estimated.
>
> > deeper analysis of simplicity
>
> We agree that there are other factors which determine the simplicity of implementing an algorithm. We removed this plot from our revised paper and now also compare other details like inference and training speed, number of parameters and training objectives, to provide a more holistic view for RL practitioners.
>
> | Method | Inference Speed | Training Speed (per batch) | Number of parameters (millions) | Number of objectives |
> |---------|---------|---------|---------|---------|
> | NF-GCBC      | 0.127 | 0.023 | 10 M | 1; end to end |
> | VQ-BET | 0.273 | 0.021 | 4.2 M | 2; pretrain a hierarchical VQ-VAE |
> | Diffusion Policy | 1.28 | 0.019 | 65.7 M | 1; end to end |
> | TarFlow | 0.273 | 0.042 | 15M | 1; end to end |
>
> NF-BC, while being competitive on all tasks, is 2, 10 times faster than VQ-BET and Diffusion policy. Unlike VQ-BET, which requires pretraining a VQ-VAE, NF-GCBC is an end to end algorithm directly optimizing for the desired likelihood. Our updated claim is that NF-BC is faster in inference, easier to implement (end to end optimization, contains 2-3 times fewer hyperparameters) and achieves competitive performance across all tasks (best success rate on two tasks and in top 2 methods on the other two tasks).
>
> We also present ablations demonstrating the robustness of NF-BC to key hyperparameters:
>
> **Success rates (higher is better):**
> | noise_std  | Push-T |
> |---------|----------|
> | 0.5     | 0.77   |
> | 0.3 | 0.79 |
> | 0.1 | 0.827|
> | 0.05 | 0.84 |
> | 0.03 | 0.873 |
> | 0.01 | 0.83 |
> | 0.0 | NAN |
>
> | number of NF blocks  | Push-T |
> |---------|----------|
> | 6   | 0.77   |
> | 14 | 0.79 |
> | 28 | 0.859 |
> | 32 | 0.85 |
> | 48  | 0.87 |
> | 54  | 0.85 |
>
> The above results show that NF-BC is robust to many values.
>
> > underperforms on a number of benchmarks
>
> Our methods exceed or match *all other baselines* on 76% (65 / 85) of tasks and is amongst the top two in seven tasks. We revised the paper to clarify that there are some small fraction of tasks where our method does not outperform the best prior methods.
>
> > comparing NF-GCBC and GCIQL
>
> Due to space, please refer to our Table 7 and OGbench (pg. 41) for GCIQL comparisons.
>
> > NFs are continuous generative models
>
> In our paper, we only focus on continuous spaces. But using NFs for large discrete spaces in RL is an exciting direction (Discrete Flows; Tran et al.). We have added this in the limitation and future work section.
>
> > Focus on online reward-based RL
>
> Due to orthogonal challenges like exploration, we chose mostly offline experiments. The toy goal reaching task described above is an online RL task, where NFs indeed perform well. Moreover, we also have experiments on online GCRL.
>
> > simulation free training
>
> Thank you for pointing this out, we have updated our paper to remove this.

---

> > ### Comment · Reviewer_aQtu · 2025-08-02
> >
> > Thank you to the authors for the response. I have a few follow-up questions and comments:
> >
> > Regarding Weaknesses 1 and 2, I feel the response did not fully address the concern. Using normalizing flows (NF) to perform VI and MLE is a well-established concept. The idea that IL and CGRL can be modeled using NF is relatively straightforward, as they are fundamentally based on MLE. As for MaxEnt RL with behavior cloning, the method in [3] can also handle this setting by simply adding a behavior cloning loss, and it also does not rely on approximation.
> >
> > Given this context, I would encourage the authors to elaborate more on the deeper connection between RL and NF—beyond just reducing approximation error. For example, how does incorporating NF influence the learning dynamics of the agent or the Q-function? Such insights would strengthen the paper’s contribution.
> >
> > Another concern is how this work positions the scope of the problem. As mentioned in Q5, RL encompasses a wide range of tasks, including those with discrete action or state spaces. The current title appears to overstate the generality of the method, whereas the experiments primarily focus on continuous control. I suggest the authors consider revising the title to better reflect the actual scope.
> >
> > Overall, I appreciate the exploration of NF in the RL setting. If the authors can address the above issues in the final version, I would be willing to increase my score to 4, as I believe the proposed method could serve as a solid and useful baseline for future work.

---

> ### Author Response · Authors · 2025-08-04
> **Reply by authors**
>
> We thank the reviewer for engaging and providing thoughtful feedback.
>
> > The current title appears to overstate the generality of the method
>
> We will update our title. More suitable titles we were thinking of are -
>
> 1) Plug-and-Play Normalizing Flows for Continuous Control.
> 2) Normalizing Flows are Capable Models for Continuous Control.
>
> Do one of these titles better reflect the scope of our experiments?
>
> > how NFs influence the dynamics of the agent?
>
> Currently, we have concrete evidence that expressiveness is an important property provided by NFs (see rebuttal above). We include below, some new results which hint towards surprising ways in which NFs affect the dynamics of RL. We believe the new results provide useful starting points to analyze how NFs affect learning dynamics.
>
> **NF policies are able to efficiently maximize complex action-value optimization landscapes**
>
> In the puzzle task, none of the baselines achieve better than 30% success rate (see Fig 6). Because solving the puzzle requires hitting specific buttons, we hypothesize that the Q function has narrow modes of good actions and a large number of bad actions.
>
> | Method | puzzle-3x3-play |
> |------------|------------|
> | NF-RLBC | 0.99 |
> | NF-AWR | 0.06 |
>
> We compare with NF-AWR which optimizes an NF policy with a sampling based optimization method AWR [4]. This does not work well despite using the same expressive NFs.
>
> **NF policies are able to maximize Q-values faster than gaussian policies in online RL**
>
> 1) Validation mean value of $Q_{SAC-NF} - Q_{SAC}$ : $-0.35, -0.01, 0.20, 0.04, 0.01, 0.24, 0.32, 0.07, 0.19, 0.36, 0.26$
> 2) Validation mean value of $(\sum_{i=t}^{H} \gamma^{i} r_i - Q_{SAC-NF}) / (\sum_{i=t}^{H} \gamma^{i} r_i)$ : $0.09$
>
> The first point shows that Q values learnt by SAC-NF are higher, indicating that the NF policy is better able to optimize the Q-function. Importantly this is not caused by overestimation of the true returns. As indicated by the second point, the Q values of SAC-NF slightly underestimate the Q values on average (by 9%).
>
> *The above two results show that apart from expressiveness, the ability of NFs to perform direct gradient based optimization is crucial in many tasks.*
>
> **NF-GCBC's stitching performance improves with training-steps**
>
> | Method | antmaze-medium-stitch | antmaze-soccer-stitch |
> |------------|------------|----------|
> | 500K | 0.532 |  0.512 |
> | 1M | 0.62 | 0.664 |
> | 1.5M | 0.68 |-|
> | 2M | 0.76 |-|
>
> Unlike TD methods, GCBC does not have an explicit mechanism for stitching. But surprisingly, we find the NF-GCBC's performance scales with training steps on tasks designed to explicitly test for stitching. A deeper investigation of this phenomena is an exciting future direction.
>
> > NF for VI and MLE is well established & the idea that IL and CGRL can be modeled using NF is relatively straightforward
>
> Even if certain ideas may appear straightforward in hindsight, there is substantial value in rigorously demonstrating their effectiveness in practice. While our experiments are based on simple ideas—which we consider a strength—the fact remains that NFs are still underutilized in RL. In contrast, diffusion models / transformers are widely used in IL [1–3], and contrastive / MLP models remain the norm for GCRL. By actually showing that NFs can often outperform these models and providing open-sourced implementations (in jax and torch), we are offering value for the community to build on.
>
> > deeper connection between RL and NF
>
> One conclusion we would like the readers of this paper to make (see line 72-74) is about rethinking how one should make RL algorithms better. Let's look at NLP. For a long time, there was lots of discussion about the right objectives in NLP: some objectives were good for sentiment classification, others for analogy making, or learning representations. However, after transformers, the field has shifted somewhat to stick with one objective (next token prediction) and instead focus on optimizing the architecture (transformers variants, mamba, etc). Our paper aims to take a first step towards doing this for RL. Of course, this paper is just a start: we're just looking at 4 problem settings. We don't claim that NFs are the best possible architecture -- it seems rather plausible that some yet-to-be-invented architecture will work even better! But the paper will hopefully help shift perceptions and show that there's this other important avenue for improving performance.
>
> We have included all the new results / text updates during the rebuttal in our paper, and we thank the reviewer for helping us do that! We hope that they address reviewer's concerns and assessment of the paper
>
> [1]  π0: A Vision-Language-Action Flow Model for General Robot Control
>
> [2] Behavior Generation with Latent Actions
>
> [3] Q-Transformer: Scalable Offline Reinforcement Learning via Autoregressive Q-Functions
>
> [4] Advantage-Weighted Regression: Simple and Scalable Off-Policy Reinforcement Learning

---

> > ### Comment · Reviewer_aQtu · 2025-08-05
> >
> > I appreciate the authors’ response. My questions were addressed. The additional rebuttal discussion provides a nice motivational example for applying normalizing flows to these RL tasks, and I recommend including that in the paper. Regarding the title, "Normalizing Flows are Capable Models for Continuous Control" may better reflect the scope of the work. I have made my final assessment accordingly.

---

> > > ### Author Response · Authors · 2025-08-05
> > > **Reply by authors**
> > >
> > > We thank the reviewer for their constant engagement and feedback, it has surely helped us improve the paper!
> > >
> > > > Normalizing Flows are Capable Models for Continuous Control
> > >
> > > We will use this title moving forward.

---

### Official Review · Reviewer_8s3P · 2025-06-26

**Clarity:** 3
**Significance:** 2
**Originality:** 3
**Rating:** 5
**Confidence:** 3

**Summary:**

This paper advocates for Normalizing Flows (NFs) as a useful building block for many RL algorithms. The core of the argument comes in two pieces: First, the work showcases that a wide variety of subproblems within RL involve maximum likelihood estimation and variational inference (Section 4) such as Imitation Learning, Offline RL, and Goal-Conditioned RL; Second, the paper presents a simple NF architecture that can be used as a modular tool across many different RL algorithms and problem settings. The design goal of the new NF architecture is characterized by the desiderata to simple, fast, and expressive. The resultant architecture combines a dense coupling network and a linear flow from prior work, along with standard layer normalization. The paper then presents a comprehensive experimental study involving a diversity of problem settings and domains, again including Imitation Learning, Offline RL, and Goal-Conditioned RL. The experimental data support the conclusion that the proposed NF architecture can act as a simple, expressive algorithmic building block across these many settings.

**Questions:**

I have two primary suggestions, along with a variety of low-level writing suggestions and comments which I include below. My two higher level suggestions are as follows:
- "Our desiderata is to build a simple and fast architecture that is expressive enough to compete and even surpass the most popular approaches across a range of RL problems.": I like this, and believe it could be emphasized even more. Additionally, you can probably weave discussion of this desiderata even more into the rest of Section 5---what alternatives did you consider? Why did you not use them? At the moment the design choices feel somewhat arbitrary (even if they are simple), so it could be useful to motivate them slightly more. similarly, the three steps around line 166-167 are quite important: it's worth spending a sentence or two expanding on them more (before the comment about the log-det).
- As mentioned in the weaknesses, I believe a slightly longer related work section will add more context to the contribution. Is it possible to provide more high-level information about the subfields discussed?

__Typos / Writing Suggestions__

Lastly, here are some low-level writing suggestions and comments:

Introduction:
- Typo: " probabilistic models which all the" --> " probabilistic models with all the"

Related work:
- The three paragraphs at present would each benefit from some synthesizing text that summarises the space; what are the open questions, what are the current limitations, and so on.

Preliminaries:
- "as dataset size N increases" --> "as the size of the dataset ($N$) increases"
- Line 85: Can you briefly mention what $Z$ is here: "... its unnormalized part $\tilde{p}(x) = p(x) \times Z$, and later state why it can be ignored?
- "jacobian matrix" --> "Jacobian matrix"
- The notation in the subscript of the Expectation in Equation 6 is slightly non-standard: $x$ is used in $z \sim p_0(x)$ and again $x = f_\theta^{-1}(z)$; the latter seems unecessary.
- Typo: " multiple RL problems settings" --> " multiple RL problem settings"
- " underlying controlled markov process" --> "underlying controlled Markov process"
- Line 126: Personal preference, but I would not index $s$ and $a$ by $t$ when mentioning the reward function here: " a reward function r(s_t, a_t)" --> " a reward function $r(s, a)$" (but I would use the time indexing on line 128 when the states and actions are inside of the sum)
- $\mathcal{D}$ is a set, but the notation $\sim \mathcal{D}$ is used---of course I assume this means there is some implicit probability mass function on $\mathcal{D}$, but this could be stated explicitly somewhere the make the notation more precise.
- Line 139: "We will define the" --> "We define the"
- Line 145: "apriori" --> "a priori"
- 'connected “coupling” network': Why the quotes?
- In the Figure 3 caption, the "(a − d)" reads as just slightly confusing, in part because the "-" is rendered as a subtraction symbol (you can replace with a-d instead of $a-d$). Or alternatively you might replace with (Subfigures a-d).
- "Simplicity is measured by the inverse of the number of hyper-parameters": This is reasonable, but could use with slightly more justification (it's the same justification as BIC and AIC---Bayesian and Akaikei Information Criterion---which provide different accounts of simplicity).
- I believe the acronym for MDP was introduced without being defined. You introduce controlled Markov processes but not MDPs.

**Ethical Concerns:**

["NO or VERY MINOR ethics concerns only"]

**Final Justification:**

This is a strong paper, with a sharp focus and a compelling experimental study: The paper effectively demonstrates that Normalizing Flows can be an important building block for many RL components.

Several poignant questions were raised during the initial reviews and rebuttals. Most notably, my two concerns throughout the discussion were regarding the motivation of various design choices made, and the scope of the claims made regarding RL, MLE, and Variational Inference. To me, these concerns have been completely addressed in the rebuttal, and I maintain a strong outlook on the work.

**Limitations:**

Yes, I believe as needed for this kind of work the authors disclose potential limitations of NFs.

**Quality:**

3

**Strengths And Weaknesses:**

__Strengths__
- [Sharp Narrative] This is a very focused paper: the whole narrative is framed around a single idea, and as such the communication of the idea is clear from start to finish.
- [Code] Implementations are provided in both Jax and PyTorch.
- [Simple Architecture] The proposed architecture is quite simple, and I imagine others in the community can quickly make use of it.
- [Broad Experimental Study] The described experimental study is diverse and comprehensive. Many baselines are used, results are thoughtfully communicated, and the strength of the claims made matches the strength of the evidence.

__Weaknesses__
- [Justification] A few design decisions could be motivated in more detail. For instance, simplicity of an algorithm is measured according to the inverse of the number of its hyperparameters. This is one choice---I would find a more faithful plot to be one that simply reports that number directly, rather than relabelling it "simplicity".
- [Clarity] There are a few areas where some added elucidation of the central ideas will help readers new to some of the ideas. For instance, MDPs are never defined, and the introduction of the coupling networks is quite quick with little added context.
- [Related Work] This is a minor one, but the discussion around the related work would benefit from a deeper discourse on the areas discussed.

---

> ### Author Rebuttal · Authors · 2025-07-30
>
> We thank the reviewer for their insightful feedback and constructive comments. It seems that the reviewer’s main feedback are regarding (1) motivations of the chosen design choices, (2) elucidation of the central ideas like coupling networks and MDPs and a more detailed related work section.
>
> To address (1), we have added new ablation experiments for all our design choices, underscoring their importance. We have also added new comparisons with two other NF architectures, showing why our proposed architecture is best suited for RL algorithms.
>
> To address (2), we have modified our paper to introduce MDPs clearly, and added a deeper discussion of coupling networks. We have also added a detailed related work section in the appendix.
>
> **Do these answers, revisions and experiments address all the reviewer's concerns?**
>
> > motivating design decisions
>
> The architectural choices we make are optimized for performance, speed, and stability with MLE and VI. Below we perform new experiments to compare our method with RealNVP [1] an old NF architecture and TarFlow [2] a SOTA NF architecture in computer vision.
>
> **Success rates (higher is better):**
> | Method  | scene-play | cube-single-play |
> |---------|---------|----------|
> | NF-GCBC      | 0.42 | 0.795 |
> | TarFlow | 0.456 |  0.594|
> | RealNVP | 0.02 |  0.04 |
>
> **Log-likelihoods (higher is better):**
> | Method  | scene-play | cube-single-play |
> |---------|---------|----------|
> | NF-GCBC | 2.756 | 2.975 |
> | TarFlow | 2.902 |  3.01 |
> | RealNVP | 1.22 |  NAN |
>
> | Method  | Inference Speed (seconds, lower is better) |
> |---------|---------|
> | NF-GCBC      | 0.127|
> | TarFlow | 0.38 |
> | RealNVP | 0.120 |
>
> From the above results, it is evident that
> 1) RealNVP is fast but unstable (likelihoods explode). Hence the combination we propose, of interleaving linear permutation flows and layernormed coupling networks is necessary for high performance.
> 2) TarFlow is able to model the distributions well and perform at par with NF-GCBC on one task, but it is three times slower than NF-GCBC. This is because in auto-regressive architectures like Tarflow (and IAF [5]), the map from action to noise can be parallelized, but the inverse map cannot. The inverse map needs to auto-regressively generate every index of the action outputs. This makes sampling and VI very slow. For tasks like offline RL or unsupervised goal sampling, where all three are required, training as well as inference are slowed down. Because coupling networks are fast for both sampling and likelihoods, we chose them over auto-regressive models.
>
> We also add new experiments ablating the inclusion of LayerNorm, a generalized permutation layer, and the sampling trick:
>
> **Success rates (higher is better):**
> | Method  | antmaze-large-navigate | scene-play | cube-single-play |
> |---------|----------|----------|----------|
> | NF -GCBC     | 0.32   | 0.42 | 0.795 |
> | without permutation flows | 0.26 | 0.04 |  0.0 |
> | without LayerNorm | 0.29 | 0.3 |  0.0 |
> | without sampling trick | 0.3 | 0.42 |  0.69 |
>
> These experiments show that above architectural modifications are crucial for performance of NFs in the context of RL. We hope that the above experiments serve as sufficient motivation for our design choices.
>
> > I would find a more faithful plot to be one that simply reports that number directly, rather than relabelling it "simplicity".
>
> We agree with the reviewer and have removed that plot from our revised paper. We now compare other details of the algorithms like inference and training speed, number of parameters (related to bayesian information criterion [1]), number of training objectives, to provide a more holistic view for RL practitioners.
>
> | Method  | Inference Speed (seconds) | Training Speed (seconds; per batch) | Number of parameters (millions) | Number of objectives |
> |---------|---------|---------|---------|---------|
> | NF-GCBC      | 0.127 | 0.023 | 10 M | 1; end to end |
> | VQ-BET | 0.273 | 0.021 | 4.23 M | 2; requires pre-training a hierarchical VQ-VAE |
> | Diffusion Policy | 1.28 | 0.019 | 65.78 M | 1; end to end |
> | TarFlow | 0.273 | 0.042 | 15.M | 1; end to end |
>
> NF-BC, while being competitive on all tasks, is 2 and 10 times faster than VQ-BET and Diffusion policy. Unlike VQ-BET, which requires pretraining a VQ-VAE, NF-GCBC is an end to end algorithm which directly optimizes for the desired likelihood. Our updated claim is that NF-BC is faster in inference, easier to implement (end to end optimization of a single loss function, contains 2-3 times less hyper-parameters ) and achieves competitive performance across all tasks (best success rate on two tasks and in top 2 methods on the other two tasks).
>
> Additionally, algorithms with fewer hyper-parameters can still be difficult to tune if their performance is sensitive to the values of those hyper-parameters. Hence, we present ablation experiments to show which hyper-parameters are important to tune and which hyper-parameters NF-BC is robust to:
>
> **Success rates (higher is better):**
> | noise_std  | Push-T |
> |---------|----------|
> | 0.5     | 0.77   |
> | 0.3 | 0.79 |
> | 0.1 | 0.827|
> | 0.05 | 0.84 |
> | 0.03 | 0.873 |
> | 0.01 | 0.83 |
> | 0.0 | NAN |
>
> **Success rates (higher is better):**
> | width of NF's coupling layers  | Push-T |
> |---------|----------|
> | 128     | 0.79   |
> | 256 | 0.82 |
> | 1024 | 0.823 |
>
> **Success rates (higher is better):**
> | number of NF blocks  | Push-T |
> |---------|----------|
> | 6   | 0.77   |
> | 14 | 0.79 |
> | 24 | 0.827 |
> | 28 | 0.859 |
> | 32 | 0.85 |
> | 40  | 0.844 |
> | 48  | 0.87 |
> | 54  | 0.85 |
>
> The above results show that NF-BC is generally robust to many values. It is important to have a non zero and small noise_std value, but many values between 0.01 - 0.1 work well.
>
> > Related work section
>
> We have added a detailed related work section in the appendix of the paper.
>
> > Typos / Writing Suggestions
>
> We appreciate all of these suggestions and have updated the paper to incorporate them.
>
> [1] Estimating the Dimension of a Model

---

> ### Comment · Reviewer_8s3P · 2025-08-04
> **Re: Rebuttal**
>
> I thank the authors for their extensive and thoughtful reply, both to my review and the other reviewers. I have read the other reviews, and maintain my positive assessment of the work.
>
> In particular I believe reviewer oKbR raised several excellent questions I would like to comment on -- first, regarding whether RL can be well-thought of as primarily MLE and VI. I agree with this question completely, and also believe that the authors' response does a good job of addressing this point. All of RL is not reducible to these two problems, so making it very clear what the scope of the proposed use of NFs in the context of RL is will sharpen the work considerably. The proposed changed paragraph accounts for this (though adding slight changes to the text throughout the paper can also help strengthen the paper).
>
> Second, clarifying why exactly NFs work well in RL. The rebuttal indicates the reasoning is due to expressivity ("The main reason why NFs perform so well is their expressivity."). While this is an important part of the story, it remains unconvincing that it is the sole reason NFs perform well. That being said, my take is that this question does not need to be fully addressed in the present paper; it is sufficient to show that NFs _can_ act as generally capable models for a variety of sub-problems within RL, which I believe the paper does, and to leave some of this explanation as room for future development (even if expressivity does yield a partial answer). Giving a completely comprehensive explanation to such a question is likely a longer-term research program. I am happy to discuss this point further.
>
> Lastly, regarding my point about motivating the design decisions more thoroughly: the added details to the plots (especially the new plot taking over for the simplicity plot, described at "We now compare other details of the algorithms") offer compelling richness to the design space. I suggest including additional text in the paper to reflect this design space, and to leave readers with a sense of why certain design choices are made (as suggested in the rebuttal).
>
> Overall I am happy with the proposed changes, and I believe the paper should be accepted, barring any other major concerns from other reviewers that I may have missed. I am happy to discuss with the other reviewers about these points.

---

> > ### Author Response · Authors · 2025-08-04
> > **Reply by authors**
> >
> > We thank the reviewer for engaging with us and providing thoughtful feedback.
> >
> > > whether RL can be well-thought of as primarily MLE and VI
> >
> > We have updated our paper with the paragraph provided in our rebuttal to reviewer oKbR. As per your suggestion, we have also updated parts of the text (figure 1 caption, start of sec 4, introduction line 23).
> >
> > > sufficient to show that NFs can act as generally capable models
> >
> > We appreciate the reviewer's thoughts. To further strengthen our paper, we have added new experiments that probe the question of what other properties of NFs that make them particularly suited in these RL sub-problems.
> >
> > Currently, we have concrete evidence that expressiveness is an important property provided by NFs (see rebuttal above or rebuttal to reviewer aQtu). We include below, some new results which hint towards surprising ways in which NFs affect the dynamics of RL. We believe the new results provide useful starting points to analyze how NFs affect learning dynamics.
> >
> > **NF policies are able to efficiently maximize complex action-value optimization landscapes**
> >
> > In the puzzle task, none of the baselines achieve better than 30% success rate (see Fig 6). Because solving the puzzle requires hitting specific buttons, we hypothesize that the Q function has narrow modes of good actions and a large number of bad actions.
> >
> > | Method | puzzle-3x3-play |
> > |------------|------------|
> > | NF-RLBC | 0.99 |
> > | NF-AWR | 0.06 |
> >
> > We compare with NF-AWR which optimizes an NF policy with a sampling based optimization method AWR [4]. This does not work well despite using the same expressive NFs.
> >
> > **NF policies are able to maximize Q-values faster than gaussian policies in online RL**
> >
> > 1) Validation mean value of $Q_{SAC-NF} - Q_{SAC}$ : $-0.35, -0.01, 0.20, 0.04, 0.01, 0.24, 0.32, 0.07, 0.19, 0.36, 0.26$
> > 2) Validation mean value of $(\sum_{i=t}^{H} \gamma^{i} r_i - Q_{SAC-NF}) / (\sum_{i=t}^{H} \gamma^{i} r_i)$ : $0.09$
> >
> > The first point shows that Q values learnt by SAC-NF are higher, indicating that the NF policy is better able to optimize the Q-function. Importantly this is not caused by overestimation of the true returns. As indicated by the second point, the Q values of SAC-NF slightly underestimate the Q values on average (by 9%).
> >
> > *The above two results show that apart from expressiveness, the ability of NFs to perform direct gradient based optimization is crucial in many tasks.*
> >
> > **NF-GCBC's stitching performance improves with training-steps**
> >
> > | Method | antmaze-medium-stitch | antmaze-soccer-stitch |
> > |------------|------------|----------|
> > | 500K | 0.532 |  0.512 |
> > | 1M | 0.62 | 0.664 |
> > | 1.5M | 0.68 |-|
> > | 2M | 0.76 |-|
> >
> > Unlike TD methods, GCBC does not have an explicit mechanism for stitching. But surprisingly, we find the NF-GCBC's performance scales with training steps on tasks designed to explicitly test for stitching. A deeper investigation of this phenomena is an exciting future direction.
> >
> > > including additional text in the paper to reflect this design space
> >
> > We have revised our paper / appendix with all of the improvements from the rebuttal. For the updated simplicity plot, we have added new text in the start of section 5 and section 6.1 results paragraph.
> >
> > We thank the reviewer for their appreciation of our work and for helping us improve our paper!

---

### Official Review · Reviewer_CtEo · 2025-06-30

**Clarity:** 4
**Significance:** 3
**Originality:** 2
**Rating:** 4
**Confidence:** 4

**Summary:**

This paper examines reinforcement learning (RL) problems from a probabilistic perspective, framing RL settings as exact likelihoods, sampling, and variational inference. The authors analyze the capabilities of Normalizing Flows (NFs) and other probabilistic methods, arguing that NFs provide the capabilities necessary for efficient RL. Methodologically, they utilize existing coupling networks and linear flows to construct NF blocks, which are integrated into RL algorithms to model the policy, Q-function, and occupancy measure under different RL settings. The paper conducts extensive experiments in imitation learning, offline RL, and goal-conditioned RL, demonstrating the effectiveness of NFs in these settings.

**Questions:**

1.What is the number of steps for the invertible transformations in the experiments? Is it related to the "blocks" hyperparameter mentioned in the appendix?

2.According to Section 4, NFs are used to model the policy network, Q-value, and occupancy measure in imitation learning, offline RL, and goal-conditioned RL, respectively. Can both the policy and Q-value learning in these scenarios be modeled using NFs?

3.What are the differences between the NF architecture chosen in this paper and those of NICE and RealNVP? Why was the coupling network chosen?

Hope the authors provide relevant explanations for mentioned weaknesses above.

**Ethical Concerns:**

["NO or VERY MINOR ethics concerns only"]

**Final Justification:**

During the rebuttal, the authors conducted additional experiments to validate and analyze the motivation behind their specific NF design choices. Moreover, they provided further analysis on how the performance of their proposed method varies with different numbers of NF blocks. They also introduced an additional metric to measure model simplicity.

My only concern is that they did not provide training curves for most settings; instead, they only reported the convergence step for their own method. This makes it difficult to assess the convergence behavior of other baselines.

Taking all of the above into consideration, I recommend a Borderline Accept for this paper.

**Limitations:**

No, refer to the second and third point in weakness.

**Paper Formatting Concerns:**

There are no major formatting issues in this paper.

**Quality:**

3

**Strengths And Weaknesses:**

Strength:

(1) This paper presents a unique perspective on the classification of existing RL problems and offers a thorough discussion of the properties of various generative models, characterized by clear logic and fluent language.

(2) The experiments in the paper are extensive and comprehensive, considering different reinforcement learning scenarios such as imitation learning, offline RL, and goal-conditioned RL, which demonstrate the effectiveness of the proposed method.

Weaknesses:

(1) This paper primarily utilizes existing NF architectures to construct its NF block. However, it lacks an analysis of the motivations for choosing coupling networks as reversible transformations.

(2) Similar to the diffusion steps in diffusion models, the steps of invertible transformations are also an important hyperparameter in NFs, affecting both model expressiveness and complexity. However, there is a lack of relevant discussion on this topic. Additionally, based on Eq. (3) and (5), it appears that this paper only performs one-step transformation, making it challenging to ensure the model's expressiveness.

(3) Measuring the model's simplicity solely by the number of hyperparameters, as shown in Figure 3, is unreasonable and insufficient. I suggest that the authors incorporate model complexity into their evaluation metrics. Furthermore, most experiments in this paper do not present training curves or complexity comparisons, making it unconvincing to claim that NFs are sample-efficient for RL.

---

> ### Author Rebuttal · Authors · 2025-07-30
>
> We thank the reviewer for their insightful feedback and constructive comments. It seems that the reviewer’s main feedback is regarding (1) more motivation behind the particular NF design choices, (2) a better measure of simplicity, and (3) more discussion of the number of NF blocks and expressiveness.
>
> To address (1), we have run new ablation experiments comparing the performance, speed and stability with two other popular NF architectures.
>
> For (2), we now additionally compare the number of model parameters (complexity), number of training objectives (end to end vs step wise) with baselines, and provide plots for hyper-parameters sensitivity of our method to give a holistic view of implementation simplicity for practitioners.
>
> For (3), we add new experiments showing how performance scales with the number of NF blocks. Additionally, we apologize for the reviewer's confusion and point that (see line 104) all NF models in the paper contained multi-step transformations.
>
> **Do these answers, revisions and experiments address all the reviewer's concerns?**
>
> > analysis of the motivations for choosing coupling networks as reversible transformations
>
> The architectural choices we make are optimized for performance, speed, and stability with MLE and VI. Below we perform new experiments to compare our method with RealNVP [1] (an early NF architecture) and TarFlow [2] (a SOTA NF architecture in computer vision).
>
> **Success rates (higher is better):**
> | Method  | scene-play | cube-single-play |
> |---------|---------|----------|
> | NF-GCBC      | 0.42 | 0.795 |
> | TarFlow | 0.456 |  0.594|
> | RealNVP | 0.02 |  0.04 |
>
> **Log-likelihoods (higher is better):**
> | Method  | scene-play | cube-single-play |
> |---------|---------|----------|
> | NF-GCBC | 2.756 | 2.975 |
> | TarFlow | 2.902 |  3.01 |
> | RealNVP | 1.22 |  NAN |
>
> | Method  | Inference Speed (lower is better) |
> |---------|---------|
> | NF-GCBC      | 0.127|
> | TarFlow | 0.38 |
> | RealNVP | 0.120 |
>
> From the above results, it is evident that
> 1) RealNVP is fast but unstable (likelihoods explode). Hence the combination we propose, of interleaving linear permutation flows and layernormed coupling networks is necessary for high performance.
> 2) TarFlow is able to model the distributions well and perform at par with NF-GCBC on one task, but it is three times slower than NF-GCBC. This is because in auto-regressive architectures like Tarflow (and IAF [5]), the map from action to noise can be parallelized, but the inverse map cannot. The inverse map needs to auto-regressively generate every index of the action outputs. This makes sampling and VI very slow. For tasks like offline RL or unsupervised goal sampling, where all three are required, training as well as inference are slowed down. Because coupling networks are fast for both sampling and likelihoods, we chose them over auto-regressive models.
>
> > What are the differences between the NF architecture chosen in this paper and those of NICE and RealNVP?
>
> Our architecture is based on RealNVP. While the RealNVP on vision tasks, our focus was on boosting performance in RL tasks. The main difference which facilitates this (see table above for new comparisons) is using a generalized permutation flow layer after every coupling network, and LayerNorm (instead of the original normalization proposed in RealNVP).
>
> > Why was the coupling network chosen?
>
> Like mentioned above (after the experimental results presented above), we chose RealNVP's coupling layers for their high speed in both sampling and likelihood estimation. This is crucial for RL tasks like offline RL or unsupervised goal sampling (where all three are required) or robot imitation learning (where controllers with low latency are preferred).
>
> > steps of invertible transformations are also an important hyperparameter in NFs
>
> Indeed, the number of steps (or number of NF blocks as we call them) are an important hyper-parameter. Below we provide how the performance of NF-BC and NF-GCBC changes with different numbers of NF blocks.
>
> **Success rates (higher is better):**
> | number of NF blocks  | Push-T |
> |---------|----------|
> | 6   | 0.77   |
> | 14 | 0.79 |
> | 24 | 0.827 |
> | 28 | 0.859 |
> | 32 | 0.85 |
> | 40  | 0.844 |
> | 48  | 0.87 |
> | 54  | 0.85 |
>
> **Success rates (higher is better):**
> | number of NF blocks  | antmaze-large-navigate | scene-play | cube-single-play |
> |---------|----------|----------|----------|
> | 6     | 0.32   | 0.42 | 0.795 |
> | 12 | 0.34 | 0.36 |  0.86 |
> | 24 | 0.36 | 0.5 |  1 |
>
> > Additionally, based on Eq. (3) and (5), it appears that this paper only performs one-step transformation, making it challenging to ensure the model's expressiveness
>
> We would like to point out that (see line 104) all NF models in the paper contained multi-step transformations. The blocks parameter is the number of NF blocks (invertible non linear transformations) stacked together to create the NF. It is mentioned in Tables 2,3,5 and A.4 for all our methods.
>
> > Measuring the model's simplicity solely by the number of hyper-parameters
>
> We agree that there exist other factors which also determine simplicity of implementing an algorithm (from a practitioner's perspective). We have removed this plot from our revised version of the paper. We now compare other details of the algorithms like inference and training speed, number of parameters (related to bayesian information criterion [3]), and number of training objectives. We believe that these new comparisons provide a more holistic view for RL practitioners.
>
> | Method  | Inference Speed (seconds) | Training Speed (seconds; per batch) | Number of parameters (millions) | Number of objectives |
> |---------|---------|---------|---------|---------|
> | NF-GCBC      | 0.127 | 0.023 | 10 M | 1; end to end |
> | VQ-BET | 0.273 | 0.021 | 4.23 M | 2; requires pre-training a hierarchical VQ-VAE |
> | Diffusion Policy | 1.28 | 0.019 | 65.78 M | 1; end to end |
> | TarFlow | 0.273 | 0.042 | 15.M | 1; end to end |
>
> NF-BC, while being competitive on all tasks, is 2 and 10 times faster than VQ-BET and Diffusion policy. Unlike VQ-BET, which requires pretraining a VQ-VAE, NF-GCBC is an end to end algorithm which directly optimizes for the desired likelihood. Our updated claim is that NF-BC is faster in inference, easier to implement (end to end optimization of a single loss function, contains 2-3 times less hyper-parameters ) and achieves competitive performance across all tasks (best success rate on two tasks and in top 2 methods on the other two tasks).
>
> Additionally, algorithms with fewer hyper-parameters can still be difficult to tune if their performance is sensitive to the values of those hyper-parameters. Hence, we present ablation experiments to show which hyper-parameter are important to tune and which hyper-parameters is NF-BC is robust to:
>
> **Success rates (higher is better):**
> | noise_std  | Push-T |
> |---------|----------|
> | 0.5     | 0.77   |
> | 0.3 | 0.79 |
> | 0.1 | 0.827|
> | 0.05 | 0.84 |
> | 0.03 | 0.873 |
> | 0.01 | 0.83 |
> | 0.0 | NAN |
>
> **Success rates (higher is better):**
> | width of NF's coupling layers  | Push-T |
> |---------|----------|
> | 128     | 0.79   |
> | 256 | 0.82 |
> | 1024 | 0.823 |
>
> **Success rates (higher is better):**
> | number of NF blocks  | Push-T |
> |---------|----------|
> | 6   | 0.77   |
> | 14 | 0.79 |
> | 24 | 0.827 |
> | 28 | 0.859 |
> | 32 | 0.85 |
> | 40  | 0.844 |
> | 48  | 0.87 |
> | 54  | 0.85 |
>
> The above results show that NF-BC is generally robust to many values. It is important to have a nonzero and small noise_std value, but many values between 0.01 - 0.1 work well.
>
> > most experiments in this paper do not present training curves or complexity comparisons, making it unconvincing to claim that NFs are sample-efficient for RL.
>
> Because most experiments have a fixed dataset, sample efficiency does not play a role. We do provide complete training curves for all online experiments (see Figure 7). Importantly, in any of the experiments, we do not make the claim that NFs are sample-efficient for RL. Nonetheless, we have added the gradient steps vs performance curves in the appendix of the revised paper.
>
> > Can both the policy and Q-value learning in these scenarios be modeled using NFs?
>
> In GCRL, NFs can indeed be used to approximate policies and Q-functions. In other offline RL they can only be used for policies. We have added a sentence in section 4 to emphasize this.
>
> [1] Density estimation using Real NVP
>
> [2] Normalizing Flows are Capable Generative Models
>
> [3] Estimating the Dimension of a Model

---

> > ### Author Response · Authors · 2025-08-04
> > **Follow-up on Revisions and Reviewer Feedback**
> >
> > Dear Reviewer,
> >
> > We have worked hard to incorporate the review feedback by running new experiments and revising the paper. Do the revisions and discussions above address your concerns? We would greatly appreciate your engagement.
> >
> > Thanks!
> >
> > The Authors

---

> > ### Comment · Reviewer_CtEo · 2025-08-05
> >
> > Thank you for your response and it addressed some of my concerns. However, I would like to follow up on a few points. The authors stated in their rebuttal that "all NF models in the paper contained multi-step transformations". If that is the case, then Eq.(3) and Eq.(5), which appear to represent one-step transformations, may be misleading or incorrect.
> >
> > Furthermore, the authors mention in Line 35 of the paper that normalizing flows enable efficient sampling. If this "efficiency" does not refer to sample efficiency, as the authors suggest in the rebuttal, then it would be helpful to clarify what is meant by "efficient", given its repeated use throughout the paper. Sample efficiency is particularly important in RL, even when working with fixed datasets [1]. I disagree with the claim that "sample efficiency does not play a role for most experiments with a fixed dataset".
> >
> > Additionally, if the authors have added gradient steps vs. performance curves in their revised version, I encourage them to provide the corresponding data at specific convergence steps in the rebuttal. Finally, I suggest clarifying in the revised paper that the number of blocks corresponds to the steps in the invertible transformation process.
> >
> > [1] Nguyen-Tang T, Arora R. On sample-efficient offline reinforcement learning: Data diversity, posterior sampling and beyond[J]. Advances in neural information processing systems, 2023, 36: 61115-61157.

---

> ### Author Response · Authors · 2025-08-05
> **Reply by authors**
>
> We thank the reviewer for engaging with us and providing thoughtful feedback. It seems the remaining feedback is regarding (1) writing clarity (introduction of NFs and efficiency) and (2) plots showing convergence gradient steps vs performance.
>
> To address (1), we have added new text to clarify the NF introduction and made it inline with how prior works introduce NFs. We also clarify what we mean by efficiency. To address (2), we have presented the desired plots in the rebuttal.
>
> **Do these answers and revisions address the reviewer's remaining concerns?**
>
> > then Eq.(3) and Eq.(5) appear to represent one-step transformations
>
> We agree with the reviewer that the way NFs are currently introduced can cause confusion. We would like to note that Eq. (3) and Eq. (5) are indeed supposed to represent the entire multi-step NF transformations. We have added new text to resolve this confusion :
>
> It is standard practice to stack multiple single step invertible mappings ($f^i_\theta$) to make the entire NF ($f_\theta$), i.e.
> $f_\theta = f^T_\theta \circ \dots \circ f^2_\theta  \circ f^1_\theta$. Each single step mapping is called an NF block. Because each $f^i_\theta$ are invertible, the entire NF $f_\theta$, which is their composition, is also invertible. The log-det of the entire NF can be estimated by summing up the log-dets of all its blocks (chain rule of derivatives):
>
> $$\log |\frac{df_\theta(x)}{dx}| = \sum_{i=1}^T \log |\frac{df^i_\theta(x)}{ df^{i-1}_\theta(x) }|,$$
>
> where $f^{0}_\theta(x) = x$.
>
> We hope the changed paragraph adds clarity. We would like to note that this is in line with how NFs are introduced in prior works [3-5].
>
> > clarify "efficient" sampling
>
> Thank you for pointing this out. By efficiency, we mean computational efficiency, which is the computational speed of sampling and training a normalizing flow. We have clarified this in the paper and have also referenced the speed comparison shared in the previous rebuttal.
>
> > provide the corresponding data at specific convergence steps in the rebuttal
>
> Below we provide the total gradient steps and the convergence gradient steps of all experiments. The convergence gradient step is the first gradient when the performance exceeds 95% of the reported performance in the paper.
>
> *Post note - For all reported success rates in the main paper, we always set the number of gradient steps as done for baselines in prior work. For example, all results in Fig 5 and Fig 6 show the success rates after exactly 1 million gradient steps. This ensure a fairer protocol for comparisons with baselines.*
>
> **BC experiments (Figure 3)**
>
> The batch size for all experiments is 256 and the dataset sizes. For more details of these datasets see Appendix A.1, page 13 of [1].
>
> | Task | convergence epochs (total epochs) |
> |------------|------------|
> | PushT   | 55 (100) |
> | Multimodal Ant | 180 (500)  |
> | UR3 BlockPush |  215 (300) |
> | Kitchen | 220 (1000) |
>
> **GCBC experiments (Figure 5)**
>
> The batch size for all experiments is 256. For more details of these datasets see section 7 of [2].
>
> | Task | convergence steps (total steps) |
> |------------|------------|
> | scene-play-v0 | 200K (1M) |
> | cube-single-play-v0 | 100K (1M)  |
> | puzzle-3x3-play-v0 |  50K (1M) |
> | antmaze-medium-stitch-v0 | 400K (1M) |
> | humanoidmaze-medium-navigate-v0 |  250K (1M) |
> | antmaze-large-navigate-v0 | 650K (1M) |
> | antsoccer-arena-navigate-v0 |  150K (1M) |
> | antsoccer-arena-stitch-v0 | 600K (1M) |
>
> **Offline RL experiments (Figure 6)**
>
> The batch size for all experiments is 256. For more details of these datasets see section 7 of [2].
>
> | Task | convergence steps (total steps) |
> |------------|------------|
> | antmaze-large-navigate-singletask-task1-v0  |  800K (1M) |
> | antsoccer-arena-navigate-singletask-task1-v0 | 200K (1M) |
> | cube-single-play-singletask-task1-v0 |  500K (1M) |
> | humanoidmaze-medium-navigate-singletask-task1-v0 | 900K (1M) |
> | puzzle-3x3-play-singletask-task1-v0 |  200K (1M) |
> | scene-play-singletask-task1-v0 | 100K (1M) |
>
> > sample efficiency does not play a role with a fixed dataset
>
> We agree with the reviewer that sample efficiency matters, especially if data is scarce. We apologize for this remark in the previous rebuttal. **We would like to point that the paper does not claim NFs are sample-efficient, nor does it make any claims about the importance of sample efficiency.** (see also clarification about efficiency^).
>
> We have included all the new results / text updates suggested during the rebuttal in our paper, and we thank the reviewer for helping us do that! We hope that they address reviewer's concerns and assessment of the paper
>
> [1] Behavior Generation with Latent Actions
>
> [2] OGBENCH: BENCHMARKING OFFLINE GOAL-CONDITIONED RL
>
> [3] Density estimation using Real NVP
>
> [4] Normalizing Flows are Capable Generative Models
>
> [5] Glow: Generative Flow with Invertible 1x1 Convolutions

---

> > ### Comment · Reviewer_CtEo · 2025-08-06
> >
> > Thank you for your further response. I suggest that the author include the training curves of both the baseline and their proposed method under various settings, as well as the explanation of normalizing flows mentioned above, in the next version of the submission. I will recommend this paper as Borderline Accept.

---

> > > ### Author Response · Authors · 2025-08-06
> > > **Reply by authors**
> > >
> > > We thank the reviewer for engaging with us and providing thoughtful feedback.
> > >
> > > > include the training curves and the explanation of normalizing flows
> > >
> > > We have included the training curves of our method in the paper. We have also included the updated explanation of normalizing flows in our paper. For the baseline scores taken from prior work which only reported the final scores, we will contact respective authors for the complete training curves.
> > >
> > > We thank the reviewer for their appreciation of our work and for helping us improve our paper!

---

### Official Review · Reviewer_oKbR · 2025-07-03

**Clarity:** 2
**Significance:** 2
**Originality:** 3
**Rating:** 4
**Confidence:** 4

**Summary:**

This paper implements Normalizing Flows(NFs) to RL problems.
The main claim is that NFs are generally capable models for RL.
The authors propose an NF architecture and algorithms using coupling networks and linear flows.
By implementing and evaluating NF to offline IL, conditional IL, offline RL, GCRL, and URL across diverse tasks, it shows a significant performance gain.

**Questions:**

A more precise context of my questions is described in the weakness section. I briefly summarize my questions below.

1. How is the performance of other NF architectures evaluated?

2. Can you provide an analysis of why NFs work well? What are the specific properties of NFs for RL performance contribution?

3. What is the detailed computational cost analysis, such as training/inference time, memory, compared to baselines?

4. What are the specific RL scenarios where NFs fail?

**Ethical Concerns:**

["NO or VERY MINOR ethics concerns only"]

**Final Justification:**

My concerns were addressed during the rebuttal. I increased my score to a 4(borderline accept).

**Limitations:**

The authors acknowledge limitations such as invertible architectures and new NF architectures, but the limitation section should be more comprehensive. My feedbacks are given in the weakness sections.

**Paper Formatting Concerns:**

No major formatting issues identified. The paper follows NeurIPS guidelines appropriately.

**Quality:**

3

**Strengths And Weaknesses:**

### Strengths

**Novel perspective for RL architecture**:
This paper opens and expands overlooked research directions on NFs for RL.

**Comprehensive study and experiment**:
This paper provides comprehensive experiments across 82 tasks in 5 RL settings.
The results show competitive performance compared to previous sota baselines with fewer parameters.
Also, this work provides practical insights about NF architecture for RL, which is noteworthy for RL practitioners to consider NF beyond diffusion and transformers.


### Weaknesses
**Overstated theoretical framework and claims**:
The paper's central claim: "RL algorithms are MLE and VI" (Section 4) is too strong. Several specific issues:

1. Claiming Q-function estimation equals occupancy distribution modeling requires very specific assumptions about reward structure (as given in references)

2. Claiming behavior cloning as MLE couldn't contain other loss function variants. (MSE loss ≠ MLE when action distribution is not Gaussian)

3. Claiming Q-learning and entropy-regularized policy optimization as variational inference is too general for what we choose for the posterior.

4. By this logic, all ML problems could be reframed as MLE+VI, which is feasible to NFs, making the argument meaningless

I hope these issues will be addressed in the rebuttal phase or further experiments.

**Lack of fundamental understanding**:

The paper lacks a fundamental understanding of why NFs perform well in RL.
Beyond "NF is good for RL", we need to find the underlying reason, such as "It's because general expressivity/specific architectural or algorithmic properties/inductive biases"
Also, failure case analysis is needed for fundamental understanding and practical implementation.
In addition, beyond the proposed architecture (RealNVP + linear flow + LayerNorm), the authors should compare other NF variants.
It is crucial to distinguish the contribution of NF properties and architecture properties.
Before that, the general claim about NFs remains unclear.

---

> ### Author Rebuttal · Authors · 2025-07-30
>
> We thank the reviewer for their insightful feedback and constructive comments. It seems that the reviewer’s main feedback is regarding (1) explaining the importance of design choices for NFs and (2) the classification of RL algorithms as VI and MLE.
>
> To address (1), we have run new experiments comparing with two other NF architectures, evaluated expressivity and the ability to do VI as the primary reason behind NF's performance, added ablations for all design choices (normalization, scaling depth, sampling trick, linear flow), computational costs and identified failure cases.
>
> Regarding (2), we have changed our claim to be more specific, and have provided counter-examples (algorithms that don’t fall into this category).
>
> **Do these answers, revisions and experiments address all the reviewer's concerns?**
>
> >  a fundamental understanding of why NFs perform well in RL. What are the specific properties of NFs for RL performance contribution?
>
> The main reason why NFs perform so well is their expressivity. Almost all datasets we use have multi-modal, noisy and high dimensional action spaces. We compare the validation likelihoods of the policies learnt by GCBC and NF-GCBC on three tasks:
>
> **Log-likelihoods (higher is better)**
> | Method  | antmaze-large-navigate | scene-play | cube-single-play |
> |---------|----------|----------|----------|
> | NF -GCBC     | -0.673   | 2.975 | 2.756 |
> | GCBC | -7.662 | -4.65 |  -4.661 |
>
> NF-GCBC is able to model the behavior policies more accurately. It is well established that higher-likelihoods are correlated with performance and generalisation in imitation learning [1,2].
>
> In offline RL experiments, the ability to do VI is important for efficient policy improvement. We add a new didactic 4 goal environment (sec 4.1 in [11]). This is an online RL experiment which consists of 4 distinct goals. We report the number of times a trained policy of any method reaches a goal, out of 20 independent rollouts:
>
> **Goal coverage (more uniform is better):**
> | Method  | goal 1 | goal 2 | goal 3 | goal 4 |
> |---------|----------|----------|----------|----------|
> | SAC     | 0   | 0 | 0 | 20 |
> | SAC-VAE | 0 | 0 |  0 | 20 |
> | SAC-NF | 8 | 2 |  7 | 3 |
>
> Clearly, SAC-NF is able to reach all 4 goals, whereas SAC-VAE, which uses a VAE state encoder and a gaussian action decoder still shows mode-seeking behavior which is a problem for reverse KL optimization. This shows that NFs which directly allow VI optimization are able to efficiently cover all the modes of the Q-function.
>
> The above two experiments suggests that expressiveness and the ability to do VI are the most important property of NFs for RL performance contribution.
>
> > How is the performance of other NF architectures evaluated? distinguish the contribution of NF properties and architecture properties
>
> In NFs, the architecture is the primary choice that affects modelling and inference. The architectural choices we make are optimized for performance, speed, and stability with MLE and VI. Below we perform new experiments to compare our method with RealNVP [3] (an old NF architecture) and TarFlow [4] (a SOTA NF architecture in computer vision).
>
> **Success rates (higher is better):**
> | Method  | scene-play | cube-single-play |
> |---------|---------|----------|
> | NF-GCBC      | 0.42 | 0.795 |
> | TarFlow | 0.456 |  0.594 |
> | RealNVP | 0.02 |  0.04 |
>
> **Log-likelihoods (higher is better):**
> | Method  | scene-play | cube-single-play |
> |---------|---------|----------|
> | NF-GCBC | 2.756 | 2.975 |
> | TarFlow | 2.902 |  3.01 |
> | RealNVP | 1.22 |  NAN |
>
> | Method  | Inference Speed (lower is better) |
> |---------|---------|
> | NF-GCBC      | 0.127|
> | TarFlow | 0.38 |
> | RealNVP | 0.120 |
>
> From the above results, it is evident that
> 1) RealNVP is fast but unstable (likelihoods explode). Hence the combination we propose – which interleaves linear permutation flows and layernormed coupling networks – is important for high performance
> 2) TarFlow is able to model the distributions well and perform at par with NF-GCBC on one task, but it is three times slower than NF-GCBC. This is because in auto-regressive architectures like Tarflow (and IAF [5]), the map from action to noise can be parallelized, but the inverse map cannot. The inverse map needs to auto-regressively generate every index of the action outputs. This makes sampling and VI very slow. For tasks like offline RL or unsupervised goal sampling, where all three are required (sampling, VI, likelihoods), training as well as inference are slowed down. Because coupling networks are fast for both sampling and likelihoods, we chose them over auto-regressive models.
>
> **We hope that this clarifies both experimentally and conceptually, the explanation behind why our design choices work well, and the properties / failure cases of different NF architectures.**
>
> > The paper's central claim: "RL algorithms are MLE and VI" (Section 4) is too strong"
>
> We agree with the reviewer that all RL algorithms are not versions of MLE and VI. We have reduced our claim and made it more specific -- *Some versions of BC, GCBC, MaxEnt RL, and GCRL are instantiations of MLE or VI*. This claim is grounded and clearly explained in equations [7,8,10,11,12]. We take the reviewer's note and have added a paragraph about counterexamples of RL algorithms that cannot be traditionally viewed as versions of MLE / VI, for example:
>
> 1) BC / GCBC with SVMs or other non probabilistic losses like 0–1 loss, hinge loss.
> 2) temporal difference learning.
> 3) State / Action abstraction learning algorithms.
> 4) Count based exploration algorithms, Planning algorithms like MCTS, algorithms based on game theory (GANs, minmax regret).
> These counterexamples show that the framework we propose in the paper does not apply to all algorithms trivially. Moreover, a rich history of work viewing RL algorithms from a probabilistic perspective [6-10], backs that the claims we make are useful for algorithmic development.
>
> > detailed computational cost analysis, such as training/inference time, memory, compared to baselines?
>
> | Method  | Inference Speed | Training Speed (seconds; per batch) | Number of parameters (millions; re: memory) |
> |---------|---------|---------|---------|
> | NF-GCBC      | 0.127 | 0.023 | 10 M |
> | VQ-BET | 0.273 | 0.021 | 4.23 M |
> | Diffusion Policy | 1.28 | 0.019 | 65.78 M |
> | TarFlow | 0.273 | 0.042 | 15.M |
>
> > specific RL scenarios where NFs fail?
>
> We have added a list of detailed failure cases of NFs in RL in the appendix. We present an outline below:
> 1) Without LayerNorm and linear permutation flows, NFs are unstable to train and sometimes result in exploding losses.
> 2) Without adding noise to the actions (noise_std hyperparameter in table 9), NFs are unstable to train and sometimes result in exploding losses.
> 3) Training with larger values of $\lambda$ (in equation 10) leads to lower performance.
>
> To further clarify the success / failure of NFs with various hyper-parameters & design choices, we run ablation experiments controlling our main design choices and hyper-parameters:
>
> | Method  | antmaze-large-navigate | scene-play | cube-single-play |
> |---------|----------|----------|----------|
> | NF -GCBC     | 0.32   | 0.42 | 0.795 |
> | without permutation flows | 0.26 | 0.04 |  0.0 |
> | without LayerNorm | 0.29 | 0.3 |  0.0 |
> | without sampling trick | 0.3 | 0.42 |  0.69 |
>
> | noise_std  | Push-T |
> |---------|----------|
> | 0.5     | 0.77   |
> | 0.3 | 0.79 |
> | 0.1 | 0.827|
> | 0.05 | 0.84 |
> | 0.03 | 0.873 |
> | 0.01 | 0.83 |
> | 0.0 | NAN |
>
> The above two tables suggest that NFs fail without permutation flows and LayerNorm, and performance decreases at very low and very high noise_std values.
>
> [1] Behavior Transformers: Cloning-K modes with one stone
>
> [2] Diffusion Policy: Visuomotor Policy Learning via Action Diffusion
>
> [3] Density estimation using Real NVP
>
> [4] Normalizing Flows are Capable Generative Models
>
> [5] Improving Variational Inference with Inverse Autoregressive Flow
>
> [6] Reinforcement Learning and Control as Probabilistic Inference: Tutorial and Review
>
> [7] What type of inference is planning?
>
> [8] General duality between optimal control and estimation
>
> [9] Robot trajectory optimization using approximate inference
>
> [10] Modeling purposeful adaptive behavior with the principle of maximum causal
> entropy
>
> [11] Maximum Entropy Reinforcement Learning via Energy-Based Normalizing Flow

---

> > ### Author Response · Authors · 2025-08-04
> > **Follow-up on Revisions and Reviewer Feedback**
> >
> > Dear Reviewer,
> >
> > We have worked hard to incorporate the review feedback by running new experiments and revising the paper. Do the revisions and discussions above address your concerns? We would greatly appreciate your engagement.
> >
> > Thanks!
> >
> > The Authors

---

> ### Comment · Reviewer_oKbR · 2025-08-07
>
> I am really thankful for the detailed response, comprehensive explanation, and hard work of the authors. My concerns are almost resolved.
> But I needed some time for careful consideration about the why question.
> While authors' comprehensive experiments give a significant explanation, they still lack essential (theoretical analysis or well-designed toy/counter experiment, etc.) understanding about 'why NFs are good?'
> It's not about expansion (of experimental task, etc.). I think it's about narrowing down/restricting to get a fundamental understanding.
>
> That's my last concern, but I think it's too harsh or too high bar for a newly-discovered sub-area.
> I believe authors will find and release the answers in camera-ready or future work.
>
> I raise my score to 4. Thanks for the great work.

---

> ### Author Response · Authors · 2025-08-08
> **Reply by authors**
>
> We thank the reviewer for their appreciation of our work and for helping us improve our paper.

---

### Decision · Program_Chairs · 2025-09-17

**Decision:**

Accept (poster)

**Comment:**

After a robust discussion, the reviewers agree that the paper makes a valuable and novel contribution, both by providing clear conceptual frameworks for categorizing RL algorithms and by exploring a rarely used model class for reinforcement learning agents. The authors should attend carefully to points that tripped up the reviewers, particularly overemphasis on the centrality of value iteration in RL, so that these questions can be anticipated and addressed for other readers.